# Orthogonal Deep Neural Networks (ODNN): Uncovering Hidden Physics in Partially Observable Systems

## Abstract

Accurately identifying the underlying physical laws in complex systems is vital for effective control and interpretation. However, many systems are governed by a combination of known physical principles and unobservable or poorly understood components. Traditional model-based methods like Kalman filters and state-space models often rely on oversimplified assumptions, while modern data-driven approaches, such as physics-informed neural networks (PINNs), can suffer from overfitting or lack theoretical guarantees in recovering true physical dynamics. We propose the Orthogonal Deep Neural Network (ODNN) architecture to address these limitations. ODNN disentangles known physical components from unobservable or poorly understood components by imposing orthogonal constraints on the deep neural network. Unlike additive regularization methods, ODNN converts the physical constraints directly into the network structure, ensuring that the DNN focuses on capturing the unknown or complex dynamics without overfitting. This novel approach leverages both explicit orthogonality (e.g., zero inner product) and implicit orthogonality (e.g., contrasting convexity, periodicity, or symmetry) between physical laws and disturbance components. Theoretically, we prove that ODNN provides strong guarantees for accurate system identification under mild orthogonality assumptions, building on the universal approximation theorem. Empirically, ODNN is evaluated across seven synthetic and real-world datasets, showcasing its ability to recover governing physical equations with high accuracy and interpretability. Our results demonstrate that ODNN offers significant advantages in terms of generalizability and robustness, making it a valuable framework for physics-based model identification in complex systems.

## 1 Introduction

Accurately identifying physical law is crucial for control and interpretation (Lu et al., 2020) in complex systems. They range from engineering fields, such as fluid dynamics (Domínguez et al., 2022), biological dynamics (Yazdani et al., 2020), and mechanical engineering (van den Bosch & van der Klauw, 2020), to computer science applications, such as image restoration (Zha et al., 2021) and audio enhancement (Ahmad et al., 2020). However, the underlying laws of many systems are governed by a combination of known physical laws and complex, often unobservable non-physical disturbances. For instance, speech signals corrupted by traffic or ambient sounds or power systems lacking information from third-party devices create challenges in accurate system identification (Basak et al., 2023). Traditional approaches to system identification include model-based methods such as Kalman filters, state-space models, and Bayesian inference, which often rely on simplified assumptions about the system's behavior (e.g., Gaussian noise). Meanwhile, modern data-driven methods, including deep learning frameworks like physics-informed neural networks (PINNs), attempt to uncover underlying physics by incorporating physical constraints into the learning process. However, these approaches often either oversimplify the unobservable dynamics or suffer from overfitting (Pillonetto et al., 2025; Petersen et al., 2019), and they lack theoretical guarantees for accurately recovering the true physical laws (Chen et al., 2021; Yuan & Weng, 2021).

Our work addresses this challenge by introducing the Orthogonal Deep Neural Network (**ODNN**) architecture, which disentangles the known physical components perfectly from unobservable or

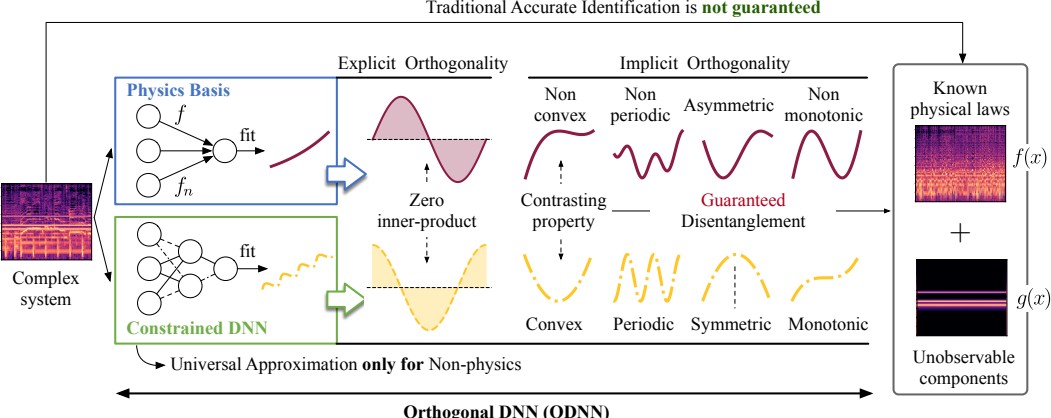

Figure 1: The proposed Orthogonal Deep Neural Network (ODNN) framework. ODNN can be applied to various real-life scenarios, covering both explicit and implicit orthogonality case.

poorly understood components for physical parameter identification. The key design lies in converting the additive physical regularization onto DNN, leading to a constrained DNN with orthogonal properties to the physical basis functions. By removing the additive physical regularization, we avoid errors due to tuning the hyperparameter. By creating a constrained DNN orthogonal to the physical basis, we avoid overfitting due to DNN's universal approximation power. We prove that a suitably restricted DNN can lead to significant loss when attempting to represent the physical equations, forcing the gradient optimizer to allocate disturbance terms to the DNN. It thus prevents overfitting and enables accurate system identification.

This idea is shown in Figure 1, where we aim to design a constrained DNN that can disentangle the physical equation $f(\cdot)$ from the non-physical disturbance $g(\cdot)$. In this paper, we present two categories of "orthogonality" that enable this disentanglement. (1) **Explicit orthogonality**: $f(\cdot)$ and $g(\cdot)$ have zero inner product. Its applications field includes image restoration (Zha et al., 2021) and audio enhancement (Ahmad et al., 2020). For example, some signals and disturbances may have approximately zero inner product following Parseval's theorem when we look into the frequency domain (Hassanzadeh & Shahrrava, 2022). For this case, we propose an orthogonal DNN that can effectively disentangle $f(\cdot)$ from $g(\cdot)$. (2) **Implicit orthogonality**: $f(\cdot)$ and $g(\cdot)$ exhibit contrasting properties, such as convexity versus non-convexity, periodicity versus non-periodicity, symmetry versus asymmetry, and monotonicity versus non-monotonicity. This case encompasses a variety of engineering systems. For example, pendulum systems exhibit symmetric dynamics contrasted by asymmetric disturbances (Sharghi & Bilgen, 2023), while bacterial growth models demonstrate non-periodic growth patterns influenced by periodic environmental factors (Egilmez et al., 2021). In power grids, non-convex power flow equations coexist with quadratic, convex disturbance terms (Xiao et al., 2024). In this case, several off-the-shelf network architectures (Raissi et al., 2019; Kiyani et al., 2023) are available to disentangle $f(\cdot)$ from $g(\cdot)$. The choices of networks are detailed in Section 3.4. For instance, the input convex neural network (ICNN) (Amos et al., 2017) is designed to output only convex functions by imposing non-negative constraints on network weights.

Theoretically, we prove that ODNN provides a formal guarantee for accurate identification under mild assumptions of orthogonality properties. Our derivation leverages the well-established universal approximation theorem (Cybenko, 1989) of neural networks: the gradient-descent optimizer will push the constrained DNN to only capture the disturbance given the orthogonality condition. This theoretical derivation offers a solid foundation for understanding and applying ODNN, as well as an analysis tool for any learning algorithm in digital twin (Pattanaik & Mohanty, 2024) structure.

We evaluate ODNN in seven synthetic and real-world datasets, covering both explicit and implicit orthogonality cases in complex systems including computer science, engineering, and critical infrastructure. We demonstrate the efficacy of ODNN in accurately recovering the governing physical equations with interpretability and generalizability. The combination of theoretical rigor and practical performance makes ODNN a valuable contribution to the field of representation learning in physics model identification.

## 2    RELATED WORK

**Traditional System Identification Methods** Classical system identification methods, such as Kalman filters (Kwasniok, 2012), state-space modeling (Haber & Verhaegen, 2020), and Bayesian inference (Huang et al., 2019), have been widely used for estimating physical systems' dynamics. These approaches have played a crucial role in early physics-based modeling by leveraging statistical properties like Gaussian noise to estimate system equations. Despite their success, such methods often struggle in dealing with complex, nonlinear systems and unobservable dynamics, limiting their applicability to real-world scenarios (Pillonetto et al., 2025). For instance, in power systems, disturbances like unmeasured third-party devices (Singh, 2021) introduce challenges that classical methods are not equipped to handle. Consequently, these approaches need to be supplemented or replaced by more flexible data-driven methods.

**Physics-Informed Neural Networks (PINNs)** Recent advances have introduced Physics-Informed Neural Networks (PINNs) (Stiasny et al., 2021), which embed physical laws directly into neural network architectures. PINNs offer improved data efficiency and interpretability by integrating known differential equations into their training processes. For instance, in the study of low-inertia systems in power grids, PINNs have been used to model frequency dynamics in the presence of significant nonlinearities and limited data (Nagel & Huber, 2022). However, while PINNs excel in situations where physical laws are well understood, they fall short when unobservable dynamics or unknown system components are present. These limitations highlight the need for models that can balance physical constraints with the discovery of unobservable dynamics.

**Neural-Symbolic Integration and Sparse System Identification** Sparse system identification methods, such as Sparse Identification of Nonlinear Dynamical Systems (SINDy) (Brunton et al., 2016) and symbolic regression techniques (Chen et al., 2021; Quade et al., 2016), aim to discover governing equations from data while minimizing the number of terms required. These approaches are powerful in their ability to capture essential dynamics but are often hampered by noise and data scarcity. Neural-symbolic integration methods have emerged to address these challenges by combining the expressiveness of deep neural networks with the interpretability of symbolic representations (Tian et al., 2021). While symbolic methods like sparse regression work well for simple systems, they struggle in complex, nonlinear scenarios, necessitating approaches like ours, which are designed to disentangle physical and non-physical components explicitly.

**Orthogonality and Disentanglement in Deep Learning** Disentangling physical components from non-physical disturbances is critical for accurate system identification and generalization. In deep learning, orthogonality has been widely adopted to disentangle factors of variation, both in latent space and network parameters (Wang et al., 2020a). Methods like Variational Autoencoders (VAEs) (Kingma, 2013) and orthogonalization of network weights aim to reduce redundancy and improve the interpretability of learned representations. However, these approaches mainly focus on disentangling features for representation learning and do not address the explicit separation of physical dynamics from disturbances. Our work builds on this by introducing Orthogonal Deep Neural Networks (ODNNs), which embed orthogonality constraints tailored for identifying physical parameters in complex, partially observable systems. This approach mitigates overfitting and enhances generalization by ensuring that physical equations are disentangled from disturbances.

## 3    METHODOLOGY

### 3.1    PROBLEM STATEMENT

Identifying the underlying physics in unobservable systems is crucial for improved interpretation and control across various domains, including fluid dynamics, power systems, and biological processes. The system can be modeled as:

$$y_i = F(\mathbf{x}_i) = \sum_{j=1}^{n} \theta_j^* f_j(\mathbf{x}_i) + g(\mathbf{x}_i), \tag{1}$$

where $(\mathbf{x}_i, y_i)$ represent the observed data, $\mathbf{x}_i \in \mathbb{R}^m$ and $y_i \in \mathbb{R}$. $f_j(\cdot)$ are known physical equations with unknown parameters $\theta_j^*$, and $g(\cdot)$ accounts for unstructured components or disturbances. Our objective is to identify the true physical parameters $\theta_j^*$, a task complicated by the interference of $g(\cdot)$, which may not follow simple noise models. The setting in Equation (1) is widely used in

complex system research, particularly in system identification (VanDerHorn & Mahadevan, 2021) and physics-informed learning (Chen et al., 2021; Huang & Wang, 2022). Its simplicity makes it easily adaptable to more intricate scenarios and enables straightforward derivation of performance guarantees, which could serve as a valuable theoretical foundation for future research.

## 3.2 CHALLENGES WITH REGULAR DEEP NEURAL NETWORKS

In complex systems, disturbances often interact with known physical dynamics in ways that are difficult to separate. When using regular Deep Neural Networks (DNNs) to approximate the disturbance components $g(\mathbf{x})$, the optimization typically aims to minimize (Gong et al., 2023):

$$\min_{\theta_j, \eta} \ \mathbb{E}_x[F(\mathbf{x}) - \sum_{j=1}^{n} \theta_j f_j(\mathbf{x}) - h_{\text{DNN}}(\mathbf{x}; \eta)]^2, \tag{2}$$

where $h_{\text{DNN}}(\mathbf{x}; \eta)$ denotes a regular DNN with network weights $\eta$ that attempts to model the disturbance components $g(\mathbf{x})$. However, due to the DNN's universal approximation capabilities, it tends to fit both the disturbance and the known physical functions $f_j(\mathbf{x})$. This results in overfitting, preventing accurate identification of the true physical parameters $\theta_j^*$.

To illustrate, consider a synthetic case where: $F(x) = \theta^* x^2 \sin(5x) + \cos(x)$ where the term $x^2 \sin(5x)$ represents known physics, and $\cos(x)$ is a disturbance. As shown in Figure 2, even though a regular DNN can minimize the error in Equation (2), it often fits both components, leading to inaccurate estimates of the physical parameter $\theta^* = 0.5$. This overfitting issue is formalized in Proposition 1.

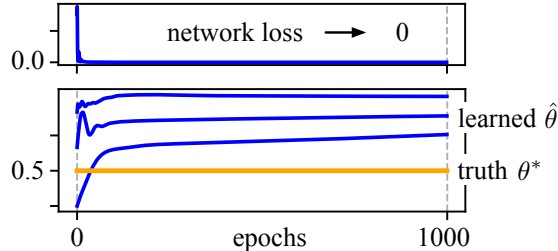

Figure 2: Overfitting issue of a regular DNN.

**Proposition 1** (Regular DNN Failure). *$\hat{\theta}_j \neq \theta_j^*, j = 1, \cdots, n$ are also minimizer of the Equation (2). Hence, a regular DNN trained via (2) is not guaranteed to converge to true parameters.*

The proof is in Appendix B.1. The overfitting issue occurs because the DNN's representational power allows it to fit both disturbance and physics, which leads to **biased** estimates of $\theta_j^*$.

## 3.3 ORTHOGONAL DEEP NEURAL NETWORK FOR GUARANTEED PHYSICS IDENTIFICATION

To overcome the overfitting issue inherent in regular DNNs, we propose the Orthogonal Deep Neural Network (ODNN) framework. ODNN is designed to enforce a **disentanglement** between the known physical dynamics and unstructured disturbances by constraining the DNN to operate in an orthogonal space relative to the physical components. The term "unstructured" is used to suggest that while the form of $g(\mathbf{x})$ is not known, certain properties of $g(\mathbf{x})$ can be inferred to enable our disentanglement of $f(\mathbf{x})$ from $g(\mathbf{x})$. This approach also aligns with the broader framework of disentangled representation learning, which seeks to develop representations that separate a system into independent components for improved interpretation and control (Wang et al., 2022).

**Motivation.** Our idea is motivated by the following reasoning. Traditionally, one common method to mitigate the overfitting issue of regular DNNs is to add a regularization term to the loss function. This approach has led to the development of various techniques within the domain of Physics-Informed Neural Networks (PINNs) (Kaheman et al., 2020), which incorporate physics-based constraints by adding penalty terms to the objective function. However, while these methods provide a useful way to include domain knowledge, they often lack formal guarantees for accurate parameter estimation as they rely on additive regularization with hyperparameter tuning. For a guaranteed estimation, we propose a novel form of regularization: rather than adding penalty terms, we directly constrain the representational capacity of the DNN. By doing so, we ensure that the DNN's universal approximation power is focused solely on the disturbance components, eventually disentangling the physics from disturbance. This constrained approach prevents overfitting and provides a more principled and theoretically sound solution for accurate system identification.

In order to formalize the requirements for effective disentanglement, we introduce Assumption 1, which outlines the conditions necessary for ODNN to separate physical and disturbance components.

This assumption ensures that the DNN can model disturbances accurately while remaining incapable of approximating the physical dynamics.

**Assumption 1** (Disentanglement between physics and disturbances). *For the physical equation $f(\cdot)$ and the disturbance equation $g(\cdot)$, the constrained DNN group $\mathcal{H}_{ODNN}$ satisfies that (1) for $\forall \varepsilon > 0$, there exists a neural network $h_{ODNN} \in \mathcal{H}_{ODNN}$ such that $\mathbb{E}_{\mathbf{x}}[g(\mathbf{x}) - h_{ODNN}(\mathbf{x})]^2 < \varepsilon$, and (2) there exists $\delta > 0$, for all neural networks $h_{ODNN} \in \mathcal{H}_{ODNN}$ we have $\mathbb{E}_{\mathbf{x}}[f(\mathbf{x}) - h_{ODNN}(\mathbf{x})]^2 \geq \delta$.*

Assumption 1 essentially guarantees that the DNN can approximate the disturbance $g(\mathbf{x})$ to any desired precision, while at the same time being unable to approximate the physical component $f(x)$. This foundational principle guides the design of ODNN, ensuring that the network focuses only on modeling the unstructured disturbances without interfering with the known physical dynamics. We introduce Assumption 1 to establish the foundation for Theorem 1, where we formally show the theoretical guarantees for accurate identification of physical dynamics.

**Theorem 1** (Accurate system identification). *For constrained DNN $\mathcal{H}_{ODNN}$ satisfying Assumption 1, any minimizer of the loss function in (2) corresponds to the correct physical parameter $\theta_j^*$.*

The proof can be found in Appendix B.2. The proof leverages the distinct loss behavior described in Assumption 1, ensuring that the gradient-based optimization process assigns the unstructured disturbance components to the constrained DNN while preserving the integrity of the governing physical equations.

**From theory to practice.** A natural question that arises is: what characteristics must a constrained DNN have to satisfy Assumption 1? Intuitively, it can be achieved by ensuring that the output space of the DNN is **orthogonal** to the functional basis of the physical components. The core idea here is to constrain the DNN to operate within a subspace that is orthogonal to $f(\mathbf{x})$. This allows for a natural separation between the known physics and the disturbances. This design fundamentally differentiates ODNN from traditional approaches, such as those in PINNs, which use additive regularizations rather than modifying the architecture itself.

To illustrate, consider an example from reinforcement learning in the Humanoid Standup environment (Brockman, 2016). In this scenario, an agent is rewarded for upward movement, which is captured by the true reward function $f(\mathbf{x})$. However, in practice, the reward signal is often corrupted by disturbances $g(\mathbf{x})$ arising from sensor drift, calibration errors, adversarial manipulation, or ground reaction force variability. Typical RL algorithms, such as Proximal Policy Optimization (Schulman et al., 2017) or Soft Actor-Critic (Haarnoja et al., 2018), assume that the reward signal is clean and reliable. When misleading factors are introduced, these algorithms struggle to distinguish between meaningful reward contributions and misleading signals, often resulting in suboptimal policies or convergence to incorrect behaviors. To handle these disturbances, ODNN is designed to disentangle the disturbances from the true reward signal. Since the true reward for upward movement is expected to be a monotonic function, the disturbance $g(\mathbf{x})$, which may introduce non-monotonic or oscillatory behavior, must be captured by a network constrained to avoid monotonic patterns. By designing a DNN that operates "orthogonally" to the monotonic characteristics of $g(\mathbf{x})$, we ensure that only the disturbances are captured, leaving the true reward function unaffected. This allows the agent to learn from a clean reward signal that accurately reflects its desired upward movement.

Based on this intuition, we name our framework the Orthogonal Deep Neural Network (**ODNN**), emphasizing its use of orthogonality to disentangle the components of the system.

### 3.4 ILLUSTRATIONS OF ORTHOGONALITY BETWEEN PHYSICS AND DISTURBANCE

This section presents practical illustrations of how orthogonality is achieved in ODNN, both through explicit orthogonality and implicit orthogonality.

- **Explicit orthogonality**. Explicit orthogonality, as a direct quantifiable relationship between two functions, indicates that the inner product between $f$ and $g$ is zero, i.e., $\langle f, g \rangle = 0$. This scenario reflects the typical situation of signal corrupted by disturbance noise in various real-life applications, such as image restoration (Zha et al., 2021) and audio enhancement (Ahmad et al., 2020). Since the signal and disturbance generally occupy distinct frequency bands, their inner product can be considered approximately zero. It follows the Parseval's theorem (Hassanzadeh & Shahrrava, 2022) that the inner product of two signals in the time domain equals the inner product

of their respective spectra, i.e., $\langle f, g \rangle = \frac{1}{2\pi} \int_{-\pi}^{\pi} F(\omega)G(\omega)d\omega \approx 0$, where $F(\omega)$ and $G(\omega)$ is the Fourier transform spectra of $f(\mathbf{x})$ and $g(\mathbf{x})$. To implement this in ODNN, we project the output of a regular DNN onto the orthogonal complement of the physical function:

$$h_{\text{ODNN}}(\mathbf{x}) = h_{\text{DNN}}(\mathbf{x}) - \frac{\langle h_{\text{DNN}}, f \rangle}{\langle f, f \rangle} f(\mathbf{x}). \tag{3}$$

We show in Corollary 1 that this design satisfies Assumption 1. The proof is in Appendix B.3.

**Corollary 1.** *The ODNN defined in Equation (3) satisfies Assumption 1 if the physical equation $f(\cdot)$ and disturbance equation $g(\cdot)$ is orthogonal.*

- **Implicit orthogonality**. Besides explicit orthogonality, we recognize another category termed implicit orthogonality. This scenario represents many engineering systems where $g(\cdot)$ and $f(\cdot)$ exhibit contrasting properties in a more qualitative sense. These contrasting properties include pairs such as convexity versus non-convexity, periodicity versus non-periodicity, symmetry versus asymmetry, and monotonicity versus non-monotonicity. For instance, physical systems typically exhibit complex, non-convex dynamics, whereas disturbances are often well-approximated by convex functions (Astolfi et al., 2021). Disturbances also tend to include periodic components, such as oscillatory noise or seasonal variations (e.g., periodic excitations in damping systems), while physical behaviors are frequently non-periodic (Spitas et al., 2020). Additional descriptions of these contrasting properties are provided in Appendix A.

  In this case, several off-the-shelf network architectures are available to disentangle $f(\mathbf{x})$ from $g(\mathbf{x})$ (Raissi et al., 2019; Kiyani et al., 2023). For example, the input convex neural network (ICNN) (Amos et al., 2017) generates convex outputs by enforcing non-negative constraints on network weights. Depending on the specific contrasting property, we choose architectures as ICNN for convexity, Hopfield network (Deng et al., 2024) for symmetry, Siamese network (Ilina et al., 2022) for periodicity, and Deep Lattice Network (You et al., 2017; Yanagisawa et al., 2022) for monotonicity. Detailed discussions on the selection of these networks and their alignment with Assumption 1 are provided in Appendix A.

**Practical contribution of ODNN.** Unlike many state-of-the-art deep learning-based approaches that rely heavily on specific patterns learned from training data, ODNN is a **self-supervised** learning framework designed to generalize effectively to real-world scenarios. Many existing models, such as DCCRN (Hu et al., 2020) and Demucs (Défossez, 2021) for audio enhancement, perform well under controlled conditions. However, they could struggle with real-time disturbances that do not match training conditions, leading to issues with overfitting and poor adaptability. ODNN, by contrast, utilizes a self-supervised learning framework that requires only knowledge of the underlying physical equations and assumes an orthogonality condition between the physical system and the disturbances. These requirements are reasonable and applicable in many practical settings.

## 4 NUMERICAL RESULTS

**Datasets for explicit orthogonality.** We consider real-world signals $f(\cdot)$ corrupted with disturbances $g(\cdot)$, where they are approximately orthogonal due to their distinct frequency characteristics. (1) **Synthetic dataset**. It allows for precise manipulation of the frequency content and ensures explicit orthogonality. Specific choices of $f(\cdot)$ and $g(\cdot)$ are shown along the analysis. (2) **Audio enhancement**. We use real-world audio signals from the Librosa package (McFee et al., 2015), mixed with environmental noise. This dataset tests ODNN's ability to disentangle overlapping but approximately orthogonal signals, showcasing its effectiveness in audio processing. (3) **Watermark removal**. Images from ImageNet (Deng et al., 2009) with human-embedded watermarks are considered. The watermark, which is visually perceptible and can be interpreted symbolically, is considered $f(\cdot)$. The host image is treated as $g(\cdot)$. The objective is to identify and remove the watermark from the host image. This dataset is discussed in Appendix C.6.

**Datasets for implicit orthogonality.** We consider real-life engineering systems where $f(\cdot)$ and $g(\cdot)$ exhibit contrasting properties covering cases of monotonicity, convexity, symmetry, and periodicity. (4) **Robotics dataset**. We consider the Humanoid Standup environment from Gym package (Brockman, 2016) which trains an agent to stand up via reinforcement learning. In this dataset, we corrupt the monotonic reward function for upward movement with non-monotonic noise, simulating disturbances that may arise from calibration errors or adversaries. (5) **Power grids dataset**.

It represents a typical operating critical infrastructure. We simulate power data $p_i$ for node $i$ using MATPOWER package (MATPOWER community, 2020) following the power flow equation $p_i = \sum_k v_i v_k (G_{ik} \cos \delta_{ik} + B_{ik} \sin \delta_{ik})$, where $v_i$ is the voltage magnitude and $\delta_{ik}$ the voltage angle difference. This equation is generally non-convex which also contains a convex form when $i = k$ and $\delta_{ik} = 0$. (6) **Heat transfer dataset.** It represents a spatially distributed dynamic system, and the goal is to identify heat sources based on temperature distribution (Loehle & Frankel, 2015). We simulate four heat sources positioned along the centerlines of the environment's edges, while an unknown source is simulated at the upper-left corner. The known heat sources create a symmetric temperature distribution, whereas the unknown source introduces asymmetry. More datasets and details can be found in Appendix C.9.

**Baselines**. The following methods are utilized as baselines. (1) The **Regular DNN** trained via Equation (2) as a direct comparison to ODNN. (2) Physics-Informed Neural Network (**PINN**) (Raissi et al., 2019). We include PINN as a baseline to demonstrate the advantage of ODNN's structural orthogonality over additive regularizations. (3) Physics-Consistent Neural Network (**PCNN**) (Li & Weng, 2021). It serves as a PINN variant that regularizes the physical model to handle partial observability. (4) Sparse Identification of Nonlinear Dynamics (**SINDy**) (Brunton et al., 2016). We use SINDy to assess how ODNN compares to a sparse regression approach. (5) Equation Learner (**EQL**) (Sahoo et al., 2018) (6) Threshold Sparse Bayesian regression (**TSBR**) (Zhang & Lin, 2018). We use TSBR to assess how ODNN compares to a Bayesian-based method.

**Implementing details**. In the ODNN approach, we choose the constrained DNN architecture based on the disturbance term's property as detailed in Section 3.4. We compile the DNN with ten layers where each with approximately twenty neurons activated by ReLU functions (Agarap, 2018). During training, we set the number of iterations $T_0 = 1000$ for sufficient training. For each iteration, we sample $n_0 = 50$ mini-batches to compute gradients for advanced searching for network parameters. We update the parameters using Adam optimizer with a learning rate of $2 \times 10^{-4}$. The experiments were conducted on a single NVIDIA GPU. A comprehensive sensitivity analysis evaluating the impact of various implicit orthogonality scenarios, different physical parameters, and constrained DNN architectures on ODNN's performance is presented in Appendix C.2, using synthetic datasets.

## 4.1 SYNTHETIC DATASET ANALYSIS FOR EXPLICIT ORTHOGONALITY CASE

We simulate 1D and 2D signals for the explicit orthogonality case by corrupting the signal with disturbances occupying close but distinct frequency bands. The orthogonality is due to $\langle f, g \rangle = \frac{1}{2\pi} \int_{-\pi}^{\pi} F(\omega) G(\omega) d\omega = 0$ (Hassanzadeh & Shahrrava, 2022). Figure 3 shows the results for several baselines. ODNN achieves a mean absolute percentage error (MAPE) of less than $0.5\%$ in identifying the physical parameters, significantly outperforming the baselines. On the contrary, PCNN and EQL employ regularizations that constrain the model but fail to achieve a globally optimal solution in terms of the MAPE of parameter estimation. More results are provided in Appendix C.1.

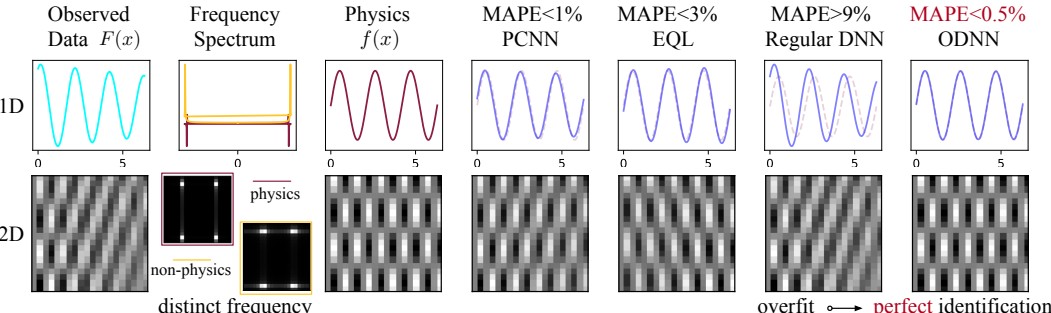

Figure 3: Performance comparison in synthetic dataset for the explicit orthogonality case.

## 4.2 AUDIO ENHANCEMENT IN AUDIO PROCESSING

Beyond the synthetic simulations, real-life signals and disturbances often exhibit overlapping frequency bands, making them not strictly orthogonal. We evaluate ODNN in these scenarios by first tackling the extraction of primary audio from noisy recordings, which remains challenging due

to complex real-world disturbances that diminish the effectiveness of traditional filtering methods (Michelsanti et al., 2021). Figure 4 presents the main audio signal $f(x)$ from the Librosa package (McFee et al., 2015), alongside the injected environmental noise and their respective frequency spectra. For simplicity, we approximate the environmental noise as a sum of sinusoidal components with a known basis. We use a music piece, Brahms Dance, and human speech from LibriSpeech as examples. Additional examples, including songs and animal sounds spanning different genres and styles, are provided in Appendix C.3. The results indicate that ODNN achieves a MAPE below $1.5\%$ across various noise simulations, effectively handling partial frequency overlap. This demonstrates ODNN's ability to exploit the tendency of the main signal and noise to occupy different but slightly overlapping frequency ranges (Makarov et al., 2020), resulting in highly effective separation.

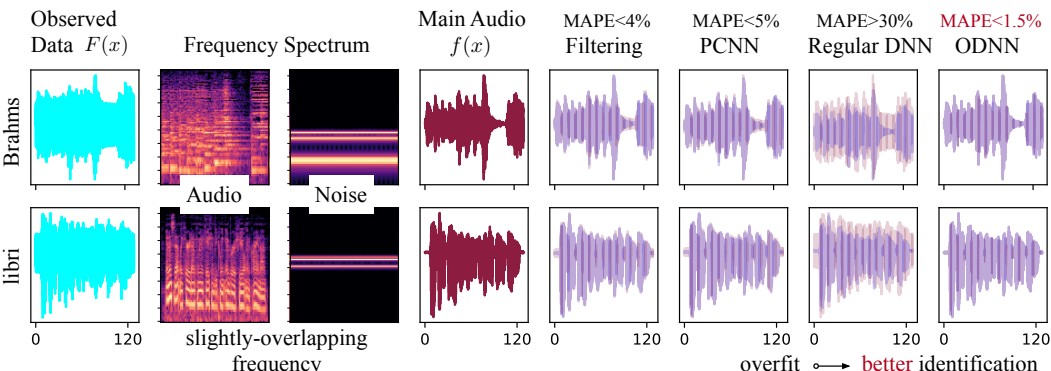

Figure 4: Audio enhancement in audio processing.

## 4.3 BALANCING VOLTAGE STABILITY AND POWER FLOW OPTIMIZATION

Besides explicit orthogonality, we also consider implicit orthogonality cases, which are representative of many engineering systems where $g(\cdot)$ and $f(\cdot)$ exhibit contrasting properties. We start from a real-industry problem: the control problem for DC system boost converter (Basati et al., 2017). For this problem, the power data is $F(V, \theta) = \sum_i P_i(V, \theta) + g(V)$, where $P_i(V, \theta) = \sum_j V_i V_j (G_{ij} \cos \theta_{ij} + B_{ij} \sin \theta_{ij})$ represents the nonconvex active power generation, where $V_i$ is the voltage magnitude of each node $i$, and $\theta_{ij} = \theta_i - \theta_j$ is the voltage phase angle difference between node $i$ and $j$. $P_i(\cdot)$ is regarded as $f_i(\cdot)$, which corresponds to the known physics basis of power flow equation, with unknown parameters $G_{ij}$ and $B_{ij}$ as the unknown conductance and susceptance between nodes $i$ and $j$. $g(V) = (V_{\text{in}})^2/(1 - D)^2$ is an additional power generation from converter $V_{\text{in}}$, which is modeled as a convex term (Basati et al., 2017) where $D$ is the duty cycle of the transistor. This experiment aims to identify the unknown parameters $G_{ij}$ and $B_{ij}$ using the distinction between non-convex $P_i(\cdot)$ and convex $g(\cdot)$. To validate our approach, we utilize the Pecan Street dataset (Street, 2024), a publicly available real-world dataset that provides detailed measurements of active power at various nodes from residential and commercial buildings. Figure 5 shows the learning trajectory of the parameters $G_{ij}$. A similar estimation accuracy exists for parameters $B_{ij}$.

The results show that multiple parameters $G_{ij}$ simultaneously converge to the true conductance values when a connection exists between buses $i$ and $j$, while they converge to zero for non-connected pairs. Then, system operators can leverage the learned parameters $G$ and $B$ to restore the network topology and recover hidden physics, enabling enhanced monitoring and operational insights for the power system.

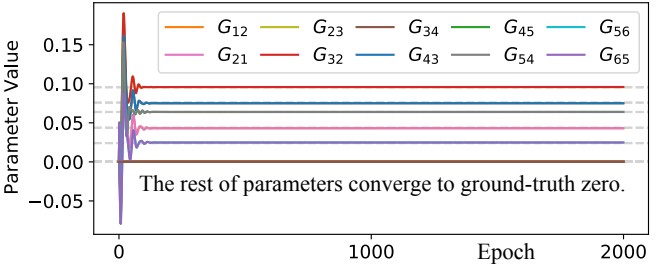

Figure 5: The learning trajectory of parameters $G_{ij}$ of line $ij$.

## 4.4 HEAT SOURCE IDENTIFICATION IN INVERSE HEAT TRANSFER PROBLEM

Besides analyzing 1D time-series data, we also consider a 2D spatially distributed dynamic system: the inverse heat transfer problem (Loehle & Frankel, 2015). In this experiment, four known heat sources are positioned along the centers of the four edge lines of the environment, while an additional unknown heat source is located at the upper-left corner. The equation governing the temperature distribution is $T_{obs}(x, y, t) = T_f(x, y, t) + T_g(x, y, t)$ where $(x, y)$ is the location and $t$ the time. For component $T_f(x, y, t)$, it is contributed from the known heat sources as $T_f(x, y, t) = \sum_{i=1}^{4} \frac{Q_i}{4\pi\alpha t} \exp\left(-\frac{(x-x_i)^2 + (y-y_i)^2}{4\alpha t}\right)$ where $Q_i, i = 1, 2, 3, 4$ represents the strength of each heat source, and $(x_i, y_i)$ are the coordinates of the four symmetric sources. More details are presented in C.8. The component $T_g(x, y, t)$ is from the unknown heat source as $T_g(x, y, t) = \frac{Q_5}{4\pi\alpha t} \exp\left(-\frac{(x-x_5)^2 + (y-y_5)^2}{4\alpha t}\right)$ where $Q_5$ is the heat strength of the noise source, located at an unknown position $(x_5, y_5)$. The heat transfer from the known sources creates a symmetric temperature distribution, while the unknown source introduces asymmetry. This distinction allows ODNN to leverage implicit orthogonality to disentangle the true heat sources from the noise. In Figure 6, the left panel shows the identified temperature heatmap, where ODNN successfully reconstructs the original four heat sources while excluding the influence of the noise source. The right panel visualizes the absolute error between the reconstructed temperature map and the ground truth. The ODNN method achieves a MAPE of less than 1%, demonstrating its high accuracy and robustness in recovering the true temperature field even in the presence of significant noise.

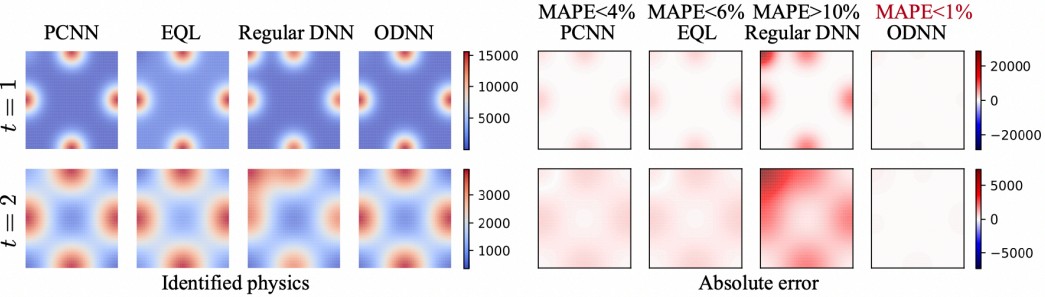

Figure 6: Temperature distribution identified in an inverse heat transfer problem.

## 4.5 COMPARISON OF PHYSICAL PARAMETER IDENTIFICATION ACROSS BASELINES

To comprehensively compare our method with baseline approaches for parameter discovery and identification across all datasets, we conducted 50 independent trials for each method, with physical parameters randomly selected within each dataset. We calculated the mean and standard deviation of the MAPE in parameter estimation, and the results are summarized in Table 1. The numerical results demonstrate that ODNN consistently outperforms traditional methods such as PCNN and Regular DNN. Unlike baselines that often suffer from overfitting, particularly in real-world conditions or require complex hyperparameter tuning to achieve reasonable accuracy, ODNN effectively manages the separation between physical dynamics and disturbances due to its inherent orthogonality constraints. This leads to significantly lower estimation errors and better robustness.

Table 1: Averaged percentage error of physical parameter estimation ± standard deviation (%).

| Dataset | SINDy | PINN | EQL | PCNN | TSBR | Regular DNN | ODNN |
|---|---|---|---|---|---|---|---|
| Synthetic system | $2.33 \pm 0.35$ | $1.15 \pm 0.22$ | $2.51 \pm 0.47$ | $0.93 \pm 0.34$ | $2.21 \pm 0.41$ | $9.21 \pm 1.55$ | $\mathbf{0.21 \pm 0.04}$ |
| Audio enhancement | $6.74 \pm 1.11$ | $4.62 \pm 0.86$ | $5.18 \pm 1.12$ | $4.04 \pm 0.59$ | $3.53 \pm 0.84$ | $17.36 \pm 1.99$ | $\mathbf{1.28 \pm 0.27}$ |
| Watermark removal | $10.56 \pm 2.03$ | $5.31 \pm 1.01$ | $8.67 \pm 1.92$ | $5.93 \pm 0.97$ | $5.28 \pm 1.42$ | $32.91 \pm 3.14$ | $\mathbf{2.83 \pm 0.49}$ |
| Robotics | $5.61 \pm 1.02$ | $3.77 \pm 0.78$ | $4.32 \pm 0.85$ | $3.65 \pm 0.45$ | $3.11 \pm 0.67$ | $19.71 \pm 2.94$ | $\mathbf{1.68 \pm 0.22}$ |
| Power grids | $7.67 \pm 0.73$ | $4.35 \pm 0.46$ | $5.11 \pm 0.87$ | $4.23 \pm 0.64$ | $2.95 \pm 0.93$ | $6.04 \pm 1.81$ | $\mathbf{0.49 \pm 0.07}$ |
| Heat transfer | $7.83 \pm 1.76$ | $3.93 \pm 0.75$ | $5.41 \pm 1.53$ | $3.28 \pm 0.82$ | $2.77 \pm 1.13$ | $11.04 \pm 2.85$ | $\mathbf{0.92 \pm 0.37}$ |
| Driven pendulum | $4.42 \pm 0.93$ | $3.87 \pm 0.85$ | $3.59 \pm 0.67$ | $3.31 \pm 0.28$ | $2.61 \pm 0.33$ | $5.18 \pm 1.45$ | $\mathbf{0.86 \pm 0.13}$ |
| Biology system | $6.72 \pm 1.25$ | $4.64 \pm 0.97$ | $8.35 \pm 1.74$ | $4.88 \pm 0.53$ | $6.92 \pm 1.21$ | $17.23 \pm 3.37$ | $\mathbf{0.84 \pm 0.15}$ |

**Comparison to more baselines.** Except from the above baselines which are general methods to learn physics, we also consider baselines that are more recent techniques designed specifically to

identify hidden physics. These baselines include a physics-informed learning of governing PDE from scarce data **PINN-SR** (Chen et al., 2021), a Bayesian spline learner **BSL** (Sun et al., 2022b) for equation discovery with quantified uncertainty, and a symbolic physics learner **SPL** (Sun et al., 2022a) leveraging Monte Carlo Tree Search to discover governing equations. Figure 7 presents the comparison results. We observe that, in most datasets, the MAPE achieved by the ODNN approach is smaller compared to the baseline methods. This improvement can likely be attributed to the distinct focus of ODNN: while the baseline methods aim to learn hidden physics by modeling the entire system as a single equation or differential equation, ODNN is specifically designed for systems that exhibit the $f + g$ structure. By leveraging the distinction between $f$ and $g$, ODNN enables a more accurate recovery of the physics parameters, ensuring robustness and precision.

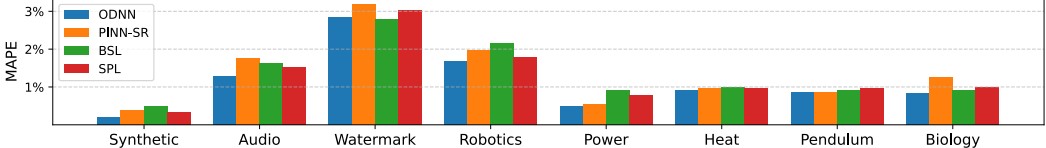

Figure 7: Mean absolute percentage error (MAPE) comparisons.

**Numerical Insights into ODNN.** ODNN outperforms baselines due to two key factors: self-supervised learning and orthogonality-based regularization. First, ODNN uses a self-supervised framework, enabling it to generalize effectively and adapt to real-time disturbances without overfitting, unlike deep learning methods (e.g., RARL (Pinto et al., 2017)) that rely on specific training patterns. Second, ODNN employs orthogonality constraints for regularization, ensuring accurate separation of physical signals from disturbances. This approach eliminates the need for complex hyperparameter tuning required by methods like PINN (Raissi et al., 2019) or PCNN (Li & Weng, 2021), providing stronger theoretical guarantees for system identification. Regarding computational efficiency, ODNN maintains similar training time as regular DNN, primarily due to its structural similarity. The main difference lies in the addition of orthogonality constraints, which introduces only a marginal computational overhead.

## 5 CONCLUSION

This paper presents a novel approach to system identification that integrates orthogonality and disentanglement into deep neural networks, enabling accurate recovery of physical equations while handling unobservable disturbances. Unlike traditional physics-informed models, which often overfit to known physical laws, our Orthogonal Deep Neural Network (ODNN) framework introduces explicit constraints that ensure the separation of physical dynamics from non-physical disturbances. By leveraging orthogonality properties within the network architecture, we demonstrate improved system interpretability, reduced overfitting, and enhanced generalizability. Through comparative analysis with state-of-the-art methods—including classical system identification, physics-informed neural networks, and hybrid neural-symbolic models—our approach outperforms these techniques in scenarios where systems are governed by partially observable or unknown dynamics. Specifically, ODNN achieves a more robust and consistent disentanglement of physical and non-physical components across various domains, ranging from power systems to mechanical and biological systems.

**Relationship to existing machine learning society.** Our method contributes to the growing body of research in self-supervised disentangled representation learning, which focuses on decomposing systems into independent, interpretable components for improved control and understanding. While previous work in disentangled representation learning, such as (Tran et al., 2017) and (Wang et al., 2022), primarily relied on heuristic methods with limited formal guarantees, our approach advances the field by providing theoretical guarantees for accurate disentanglement. This is achieved through the use of orthogonality between known physical laws and unstructured disturbances. Furthermore, our approach operates in a self-supervised setting, requiring only known physical equations, making it robust for real-world applications without needing explicit label information.

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

## A  Specific Illustrations of Implicit Orthogonality Case

Besides explicit orthogonality, we recognize another category termed implicit orthogonality. This scenario represents many engineering systems where $g(\cdot)$ and $f(\cdot)$ exhibit contrasting properties in a more qualitative sense. These contrasting properties include pairs such as convexity versus non-convexity, periodicity versus non-periodicity, symmetry versus asymmetry, and monotonicity versus non-monotonicity.

For instance, physical systems typically exhibit complex, non-convex dynamics, whereas disturbances are often well-approximated by convex functions (Astolfi et al., 2021). Disturbances also tend to include periodic components, such as oscillatory noise or seasonal variations (e.g., periodic excitations in damping systems), while physical behaviors are frequently non-periodic (Spitas et al., 2020). Similarly, physical systems are often asymmetric due to material properties or boundary conditions, whereas disturbances generally exhibit symmetric characteristics. Furthermore, while physical systems may involve complex, non-monotonic relationships, disturbances are often approximated effectively by monotonic functions.

- Convex v.s. Non-Convex. While physical systems often exhibit complex, non-convex behaviors, the characteristics of disturbances frequently lend themselves to approximation by convex functions (Astolfi et al., 2021). For instance, disturbances can often be simplified as square functions (Astolfi et al., 2021). The sensor noise in power systems are often modeled as Gaussian variables, whose square are then utilized in security analysis (Xiao et al., 2024). To capture exclusively convex functions, we leverage the off-the-shelf input convex neural network (ICNN) architecture (Amos et al., 2017), which has demonstrated significant success in various inference tasks related to convex optimization. ICNNs are designed to output convex functions by imposing non-negative constraints on network weights. As demonstrated in (Chen et al., 2020), ICNNs satisfy the universal approximation theorem for convex functions, ensuring they adhere to the Assumption 1.

- Periodic v.s. Non-Periodic. Disturbances frequently contain periodic components, while physical systems often exhibit non-periodic behaviors. For instance, oscillatory noise, such as periodic excitation in the damping system or seasonal variations in environmental factors, can be effectively modeled using sinusoidal functions (Spitas et al., 2020). To exclusively capture periodic functions, we employ Hopfield networks (Deng et al., 2024), capable of exhibiting periodic attractor states through weight sharing. These networks produce periodic outputs upon convergence.

- Symmetric v.s. Asymmetric. While physical systems often exhibit asymmetric behaviors due to factors such as material properties or boundary conditions, disturbances frequently possess symmetric characteristics. For example, random noise, a common disturbance, is often symmetrically distributed around zero. Symmetric functions, such as Gaussian distributions, are commonly used to model these types of disturbances. To capture exclusively symmetric functions, we leverage the off-the-shelf Siamese networks (Ilina et al., 2022). By adapting this architecture to induce symmetry, we aim to create a model that specifically captures symmetric disturbances $g(\cdot)$.

- Monotonic v.s. Non-Monotonic. While physical systems often exhibit complex and non-monotonic relationships between variables, disturbances can frequently be approximated by monotonic functions. For instance, gradual changes in environmental conditions or systematic measurement errors might introduce monotonic trends into the data. To exclusively capture symmetric functions, we leverage Deep Lattice Networks (You et al., 2017; Yanagisawa et al., 2022), which enforce monotonicity w.r.t. specified inputs through alternating layers of linear embeddings and lattice ensembles.

# B DETAILED PROOFS

## B.1 PROOF OF PROPOSITION B.1

**Proposition 1** (Regular DNN Failure). *$\hat{\theta}_j \neq \theta_j^*, j = 1, \cdots, n$ are also minimizer of the Equation (2). Hence, a regular DNN trained via (2) is not guaranteed to converge to true parameters.*

*Proof.* Since $h_{\text{DNN}}(\mathbf{x}; \eta)$ is a regular deep neural network, it has well-established universal approximation theorem (Cybenko, 1989). Hence, for an arbitrary network loss $\varepsilon > 0$, there exists network weights $\eta^*$ such that

$$\mathbb{E}_x \left[ \hat{F}(x) - h_{\text{DNN}}(\mathbf{x}; \eta^*) \right]^2 < \varepsilon,$$

for the continuous function $\hat{F}(x) = \sum_{j=1}^n (\theta_j^* - \hat{\theta}_j) f_j(x) + g(x)$. Substituting this DNN $h_{\text{DNN}}(\mathbf{x}; \eta^*)$ into Equation (2), we have

$$\mathbb{E}_x \left[ F(x) - \sum_{j=1}^n \hat{\theta}_j f_j(\mathbf{x}) - h_{\text{DNN}}(\mathbf{x}; \eta^*) \right]^2 < \varepsilon,$$

which indicates that $\hat{\theta}_j \neq \theta_j^*, j = 1, \cdots, n$ are also minimizer of the Equation (2). $\square$

## B.2 PROOF OF THEOREM 1

**Theorem 1** (Accurate system identification). *For constrained DNN group $\mathcal{H}_{ODNN}$ satisfying Assumption 1, such network trained via Equation (2) can learn the physical parameter $\theta_j$ correctly.*

*Proof.* We show that $\hat{\theta}_j = \theta_j^*, j = 1, \cdots, n$ are the only minimizer of Equation (2) if we utilize the constrained DNN group $\mathcal{H}_{ODNN}$ satisfying Assumption 1. In fact, suppose for some $j = j_0$ it satisfies $\hat{\theta}_{j_0} \neq \theta_{j_0}^*$, and $\hat{\theta}_j = \theta_j^*, j = 1, \cdots, n$ is also a minimizer of Equation (2). Then, it holds that for an arbitrary network loss $\varepsilon > 0$, there exists a neural network $h_{\text{ODNN}} \in \mathcal{H}_{ODNN}$ such that

$$\mathbb{E}_{\mathbf{x}} \left[ \sum_{j=1}^n \theta_j^* f_j(\mathbf{x}) + g(\mathbf{x}) - \sum_{j=1}^n \hat{\theta}_j f_j(\mathbf{x}) - h_{\text{ODNN}}(\mathbf{x}) \right]^2 < \varepsilon. \tag{4}$$

Then, we note that for this specific $\varepsilon > 0$, there exists a neural network $h_{\text{ODNN}}^g \in \mathcal{H}_{ODNN}$ such that $\mathbb{E}_{\mathbf{x}}[g(\mathbf{x}) - h_{\text{ODNN}}^g(\mathbf{x})]^2 < \varepsilon$. Hence, we have

$$\mathbb{E}_x \left[ \sum_{j=1}^n (\theta_j^* - \hat{\theta}_j) f_j(x) - (h_{\text{ODNN}}(x) - h_{\text{ODNN}}^g(x)) \right]^2 \tag{5}$$

$$< \mathbb{E}_{\mathbf{x}} \left[ \sum_{j=1}^n \theta_j^* f_j(\mathbf{x}) + g(\mathbf{x}) - \sum_{j=1}^n \hat{\theta}_j f_j(\mathbf{x}) - h_{\text{ODNN}}(\mathbf{x}) \right]^2 + \mathbb{E}_{\mathbf{x}}[g(\mathbf{x}) - h_{\text{ODNN}}^g(\mathbf{x})]^2 \tag{6}$$

$$< 2\varepsilon, \tag{7}$$

which contradicts the condition in Assumption 1: there exists $\delta > 0$, for all neural networks $h_{\text{ODNN}} \in \mathcal{H}_{ODNN}$ we have $\mathbb{E}_{\mathbf{x}}[f(\mathbf{x}) - h_{\text{ODNN}}(\mathbf{x})]^2 \geq \delta$. $\square$

## B.3 PROOF OF COROLLARY 1

**Corollary 1.** *The ODNN defined in Equation (3) satisfies Assumption 1 if the physical equation $f(\cdot)$ and disturbance equation $g(\cdot)$ is orthogonal.*

*Proof.* We first show that the ODNN defined in (3) always outputs functions that are orthogonal to the physical equation $f(\cdot)$, i.e., $\langle h_{\text{ODNN}}, f \rangle = 0$:

$$\langle h_{\text{ODNN}}, f \rangle = \int_{x_0}^{x_1} f(\mathbf{x}) \cdot \left( h_{\text{DNN}}(\mathbf{x}) - f(\mathbf{x}) \frac{\int_{x_0}^{x_1} h_{\text{DNN}}(\mathbf{x}) \cdot f(\mathbf{x}) dx}{\int_{x_0}^{x_1} f(\mathbf{x}) \cdot f(\mathbf{x}) dx} \right) dx = 0. \tag{8}$$

Then we show there exists $\delta > 0$, for all neural networks $h_{\text{ODNN}} \in \mathcal{H}_{\text{ODNN}}$ we have $\mathbb{E}_{\mathbf{x}}[f(\mathbf{x}) - h_{\text{ODNN}}(\mathbf{x})]^2 \geq \delta$. In fact, suppose the opposite is true, for arbitrary network loss $\varepsilon > 0$ there exists a neural networks $h_{\text{ODNN}} \in \mathcal{H}_{\text{ODNN}}$ we have $\mathbb{E}_{\mathbf{x}}[f(\mathbf{x}) - h_{\text{ODNN}}(\mathbf{x})]^2 < \varepsilon$. It contradicts the conclusion that $\langle h_{\text{ODNN}}, f \rangle = 0$.

Specifically, assume that for an arbitrary network loss $\varepsilon > 0$, there exists a neural network $h_{\text{ODNN}} \in \mathcal{H}_{\text{ODNN}}$ such that the approximation error is bounded by $\varepsilon$:

$$\mathbb{E}_{\mathbf{x}}\left[(f(\mathbf{x}) - h_{\text{ODNN}}(\mathbf{x}))^2\right] < \varepsilon. \tag{9}$$

This implies that the neural network $h_{\text{ODNN}}$ can approximate the function $f$ arbitrarily well, i.e., the difference between $f$ and $h_{\text{ODNN}}$ can be made arbitrarily small in the mean squared error sense. Now consider the condition that $h_{\text{ODNN}}$ and $f$ are orthogonal:

$$\langle h_{\text{ODNN}}, f \rangle = 0. \tag{10}$$

The inner product being zero implies that the functions $h_{\text{ODNN}}$ and $f$ are orthogonal in the Hilbert space sense, meaning that they are linearly independent, and there is no overlap in their representation. However, if $h_{\text{ODNN}}$ can approximate $f(\mathbf{x})$ to an arbitrary degree of accuracy, it suggests that $h_{\text{ODNN}}(\mathbf{x})$ must contain components that are aligned with $f(\mathbf{x})$. This alignment contradicts the orthogonality condition $\langle h_{\text{ODNN}}, f \rangle = 0$. $\qquad \square$

## C MORE EXPERIMENTS

### C.1 SYNTHETIC DATASET ANALYSIS FOR EXPLICIT ORTHOGONALITY CASE

We simulate both 1D and 2D signals for explicit orthogonality case by corrupting the signal with disturbances occupying distinct frequency bands. Mathematically, this orthogonality is due to $\langle f, g \rangle = \frac{1}{2\pi} \int_{-\pi}^{\pi} F(\omega) G(\omega) d\omega = 0$ based on Parseval's theorem (Hassanzadeh & Shahrrava, 2022), where $F(\omega)$ and $G(\omega)$ are the Fourier transform spectra of $f(\mathbf{x})$ and $g(\mathbf{x})$. Figure 8 shows the identification results for several baseline models. In particular, the second and fourth rows depict scenarios with close but distinct frequencies. The results indicate that the ODNN achieves a mean absolute percentage error (MAPE) of less than 0.5% in identifying the physical parameters, significantly outperforming the baseline methods. On the contrary, PCNN and EQL employ regularization techniques that constrain the model but fail to achieve a globally optimal solution.

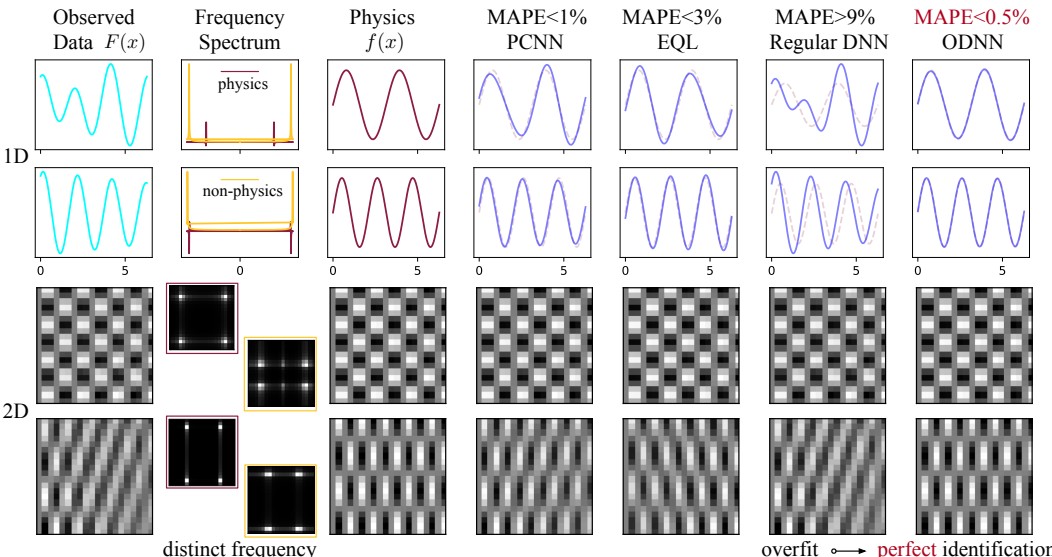

Figure 8: Performance comparison in synthetic dataset for the explicit orthogonality case.

## C.2 Synthetic Dataset Analysis for Implicit Orthogonality Case

We also consider four synthetic datasets corresponding to implicit orthogonality case. Each row in the left-half of Figure 9 shows the observed system $F(\mathbf{x})$, its associated physics equation $f(\mathbf{x})$ and the disturbance term $g(\mathbf{x})$. For example, the first row represents $F(x) = x^2 + \sin(5x)$ and our goal is to identify the physical parameter 1 corresponding to the non-convex physics basis $\sin(5x)$, while hoping that DNN will only approximate the convex part of $x^2$. Likewise, the second row represents $F(x) = \sin(5x) + x\sin(5x)$ with the goal to identify the parameter 1 before the non-periodic physics basis $x\sin(5x)$, while hoping that DNN will only capture the periodic part of $\sin(5x)$. The third row models $F(x) = \cos(5x) + x^2\sin(5x)$, aiming to identify the coefficient of $x^2\sin(5x)$ while isolating the symmetric component $\cos(5x)$. The fourth row models $F(x) = \exp(x) + \sin(5x)$, aiming to identify the coefficient of $\sin(5x)$ while isolating the monotonic component $\exp(x)$.

In the left-half of Figure 9, the right-hand side of the black rectangular area shows that ODNN recovers the physics basis perfectly with a superposition while the regular DNN tends to overfit this physics function (see Proposition 1).

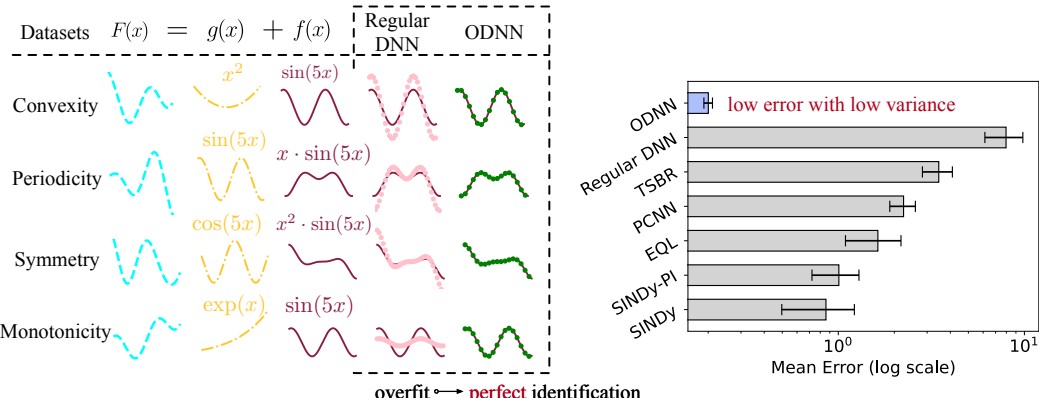

Figure 9: Left: Comparison of Regular DNN and ODNN in the implicit orthogonality case. Right: Averaged percentage error (log-scaled) of parameter estimation in synthetic datasets.

In addition to the specific example above, we calculate the averaged identification error across various parameters choices. The results in the right-half of Figure 9 shows that the ODNN leads to significantly lower averaged error and reduced variance, compared to alternative approaches.

## C.3 Speech Enhancement in Audio Processing

Beyond the synthetic simulations, real-life signals and disturbances often exhibit overlapping frequency bands, making them not strictly orthogonal. We evaluate ODNN in these scenarios by first tackling the extraction of primary audio from noisy recordings, which remains challenging due to complex real-world disturbances that diminish the effectiveness of traditional filtering methods (Michelsanti et al., 2021). For state-of-the-art deep learning-based methods, such as DCCRN (Hu et al., 2020) or Demucs (Défossez, 2021), they often rely heavily on learning distinct patterns from training data, which may not generalize well when faced with real-time disturbances that change their frequency characteristics unpredictably. Such models struggle with adaptive noise that does not conform to a fixed spectral pattern, leading to residual noise or degradation of the target signal.

Figure 10 presents the main audio signal $f(x)$ from the Librosa package (McFee et al., 2015), alongside the injected environmental noise and their respective frequency spectra. For simplicity, we approximate the environmental noise as a sum of sinusoidal components with a known basis. To ensure a comprehensive evaluation of our audio processing methods, we have carefully selected a diverse dataset encompassing a wide range of audio sources. These include classical music pieces like "Brahms - Hungarian Dance #5" and "Tchaikovsky - Dance of the Sugar Plum Fairy", popular songs like "Karissa Hobbs - Let's Go Fishin", animal sounds like "Humpback whale song" and "Bird Whistling Robin" and synthesized music pieces like "Setuniman - Sweet Waltz" and "Kevin Macleod - Vibe Ace". Additionally, we have included LibriSpeech examples to represent spoken

language in various genres and styles. This diverse dataset allows us to assess the performance of our methods on a variety of audio signals, ensuring that our models are robust and capable of handling different types of audio content. These examples are shown in Figure 10.

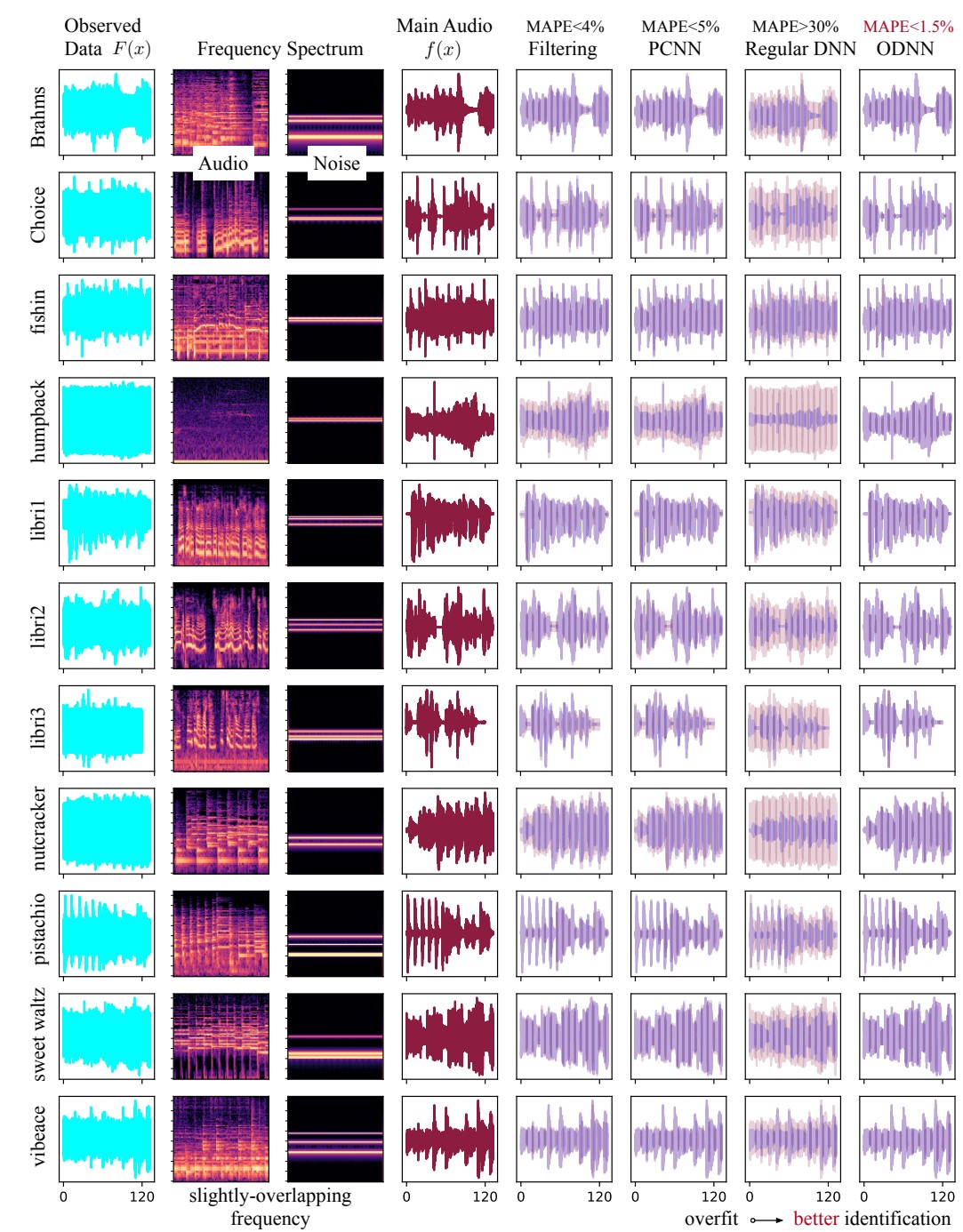

Figure 10: Audio enhancement in audio processing.

Overall, the identification results indicate that ODNN achieves a MAPE below 1.5% across various noise simulations, effectively handling partial frequency overlap. Despite partial frequency overlap, ODNN leverages the general tendency of the main signal and noise to occupy different frequency ranges (Makarov et al., 2020), allowing for an effective and near-optimal separation.

## C.4 ROBUST REWARD FUNCTION LEARNING IN ROBOTICS DATASET

For implicit orthogonality case, we consider a robotic control problem: the Humanoid Standup environment from OpenAI Gym package (Brockman, 2016). The objective is to train a robot agent to stand up from a seated position via reinforcement learning (RL). In practical RL, the reward function can be susceptible to disturbances from environmental variability or adversarial actions (Ilahi et al., 2021), complicating the development of robust RL methods (Wang et al., 2020b). In Humanoid dataset, the true reward for upward movement is $\frac{z}{\Delta t}$, where $z$ denotes the post-action z-coordinate and $\Delta t$ is the frame time. To evaluate robustness, we introduce symmetric noise $\sin(z)$ into the inherently asymmetric reward function. In Figure 11, we utilize PCNN, standard DNN, and ODNN to extract the true reward function. Based on the learned reward function, we apply Deep Q-learning (DQL) (Muzio et al., 2022) and compare them to Robust Adversarial Reinforcement Learning (RARL) (Pinto et al., 2017).

The results indicate that ODNN achieves the highest and most robust reward function at convergence. ODNN successfully enables the robot agent to stand upright, whereas other methods display deviations from the standing position due to the corrupted reward function.

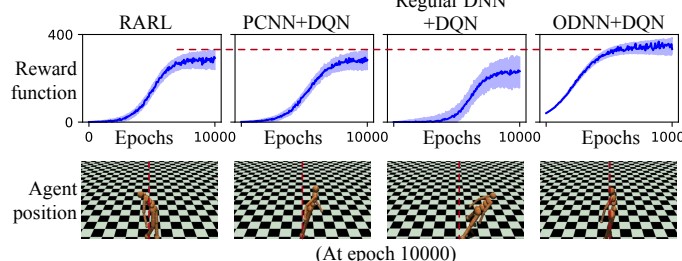

Figure 11: Robust reward function learning in robotics dataset.

### C.5 RECONSTRUCTING OSCILLATOR DYNAMICS WITH OBSERVED DATA

We also consider a real-life temporal dynamic system: the damped harmonic oscillator. In this experiment, the oscillator is initialized with an unknown velocity, and the simulated dataset is publicly available in (Cici118, 2023). The motion of the oscillator is governed by the equation: $mx''(t) + cx'(t) + kx(t) = 0$, where $x(t)$ is the displacement at time $t$, $m$ is the mass, $c$ is the damping coefficient, and $k$ is the spring constant. The dataset provides the parameters $m$, $c$, and $k$, which satisfy the underdamped condition ($c^2 < 4mk$). Under this condition, the displacement follows the solution: $x(t) = e^{-\gamma t} \left( A \cos(\omega t) + B \sin(\omega t) \right)$, where $\gamma = \frac{c}{2m}$ and $\omega = \sqrt{\frac{k}{m} - \gamma^2}$. The coefficient $A = x(0)$ is determined from the initial displacement, while $B$, related to the initial velocity, is given by $B = \frac{x'(0) + \gamma A}{\omega}$. Since the initial velocity is unknown, $B$ cannot be directly determined from the dataset.

Except from the exponential decay envelope $e^{-\gamma t}$ due to damping, the contributions from the known displacement generate predictable oscillatory behavior, which we regard as $f = A \cos(\omega t)$. The unknown velocity introduces variability $B \sin(\omega t)$ into the system, which we regard as $g$. Since we known $f$ and $g$ are distinct in their symmetry axes, it enables ODNN to leverage implicit orthogonality to disentangle the contributions of the known displacement from the unmodeled dynamics. Figure 12 illustrates the results. The left panel compares the predicted displacement of the oscillator by ODNN against the ground truth, demonstrating that ODNN effectively captures the dynamics of the system with high accuracy. The right panel shows the learning trajectory of the unknown parameter $B$, which converges to its true value, independently validated by the acceleration data (not used during training). These results highlight the capability of ODNN to recover the true motion dynamics of the oscillator, accurately disentangling the system's parameters and underlying behaviors.

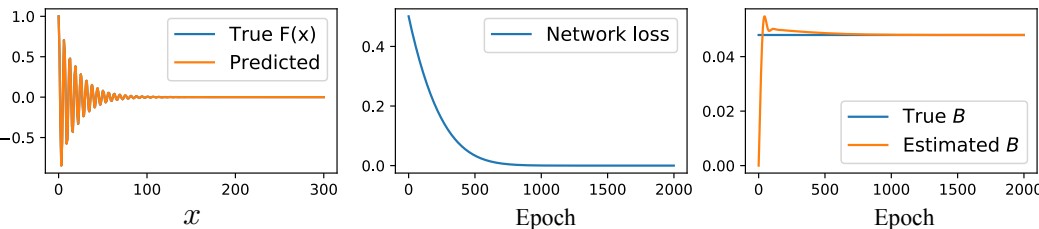

Figure 12: (Left) The predicted solution of the oscillator dataset compared to truth. (Middle) The ODNN loss function against epoches. (Right) The learning trajectory of the unknown parameter $B$.

## C.6 Watermark Removal in Image Processing

For the watermark removal experiment, we use images from ImageNet (Deng et al., 2009) with human-embedded watermarks. The watermark, visually perceptible and symbolically interpretable, serves as the target signal $f(\cdot)$ to be identified, while the host image is treated as the disturbance $g(\cdot)$. The objective is to accurately identify and remove the watermark while preserving the integrity of the host image. Users were given two options for specifying the watermark content: direct text input or manual area selection. This information was used to generate a reference watermark image, serving as the symbolic target $f(\cdot)$ for our ODNN model, as illustrated in Figure 13.

Watermark removal is traditionally challenging due to the significant overlap between watermark features and underlying image content, which often makes conventional methods prone to either over-removing host content or leaving residual watermark traces. Our ODNN approach is designed to overcome these challenges by leveraging orthogonality. Specifically, watermarks and host images tend to exhibit different frequency characteristics: watermarks are often designed with repetitive patterns or high-contrast features that manifest prominently in specific frequency bands, whereas natural images typically contain broader, non-repetitive frequency content (Gonzalez & Woods, 2008). This approximate orthogonality allows ODNN to effectively disentangle the watermark from the underlying image.

Experimental results demonstrate that ODNN can achieve high accuracy in identifying and removing watermarks, preserving the underlying image's quality even in complex cases. Moreover, our method is computationally efficient, allowing for real-time applications. The success of ODNN in watermark removal showcases its broader applicability to tasks involving symbolic versus non-symbolic content separation, which aligns well with ICLR's interest in advancing representation learning methods that incorporate disentanglement and interpretability.

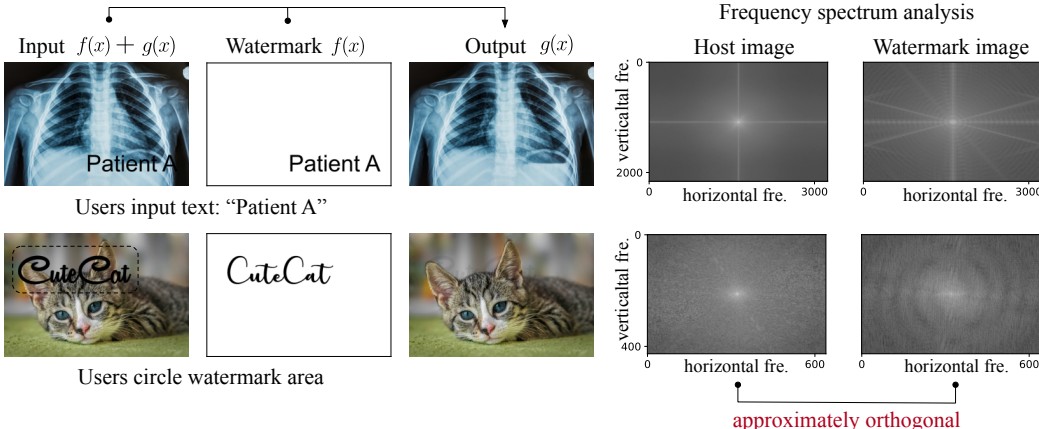

Figure 13: Watermark removal in image processing.

## C.7 LINE PARAMETERS IDENTIFICATION IN POWER GRID

For operating critical infrastructure, we use power system as an example. The power grid delivers electric power to end users and represents a typical cyber-physical system. For each node $i$, its active power $p_i$ is determined by power flow equations (Li et al., 2021): $p_i = \sum_{k=1}^{|K|} v_i v_k (G_{ik} \cos \delta_{ik} + B_{ik} \sin \delta_{ik})$, where $v_i$ is the voltage magnitude and $\delta_{ik}$ the voltage angle difference between node $i$ and $k$. $G_{ik}$ and $B_{ik}$ represent the physical parameters of line $ik$ which remain unknown. The power flow equation is non-convex which also contains a convex form, when $i = k$ and $\delta_{ik} = 0$, thereby satisfying the "contrasting property" requirement as convexity. The data is simulated in IEEE 4-, 9-, 14-, 18-, and 123-bus systems. The performances are similar, so we choose 18-bus system as an example, shown on the left of Figure 14. We use MATPOWER (MATPOWER community, 2020) and real residential data from Duquesne Light Company (Cook et al., 2021; 2022) for simulating the data on partial topology/parameter recovery. Such data is shown on the top right of Figure 14.

We denote line parameters as conductance $G_{ik}$ and susceptance $B_{ik}$ for line $ik$. They are often unknown in distribution grids necessitating an estimation (Cook et al., 2022). But, the measurements are quite limited in many distribution grids, e.g., residential ones. Figure 14 illustrates topology parameter estimation results for IEEE 18-node system with measurement from part of the network. The bottom right tables in Figure 14 compare the true and estimated physical parameters in matrices. The achieved mean absolute percentage error is less than $0.5\%$. The benchmark methods have an average error to be 10 times more, highlighting the effectiveness and theoretical soundness of our approach in handling partially observable systems.

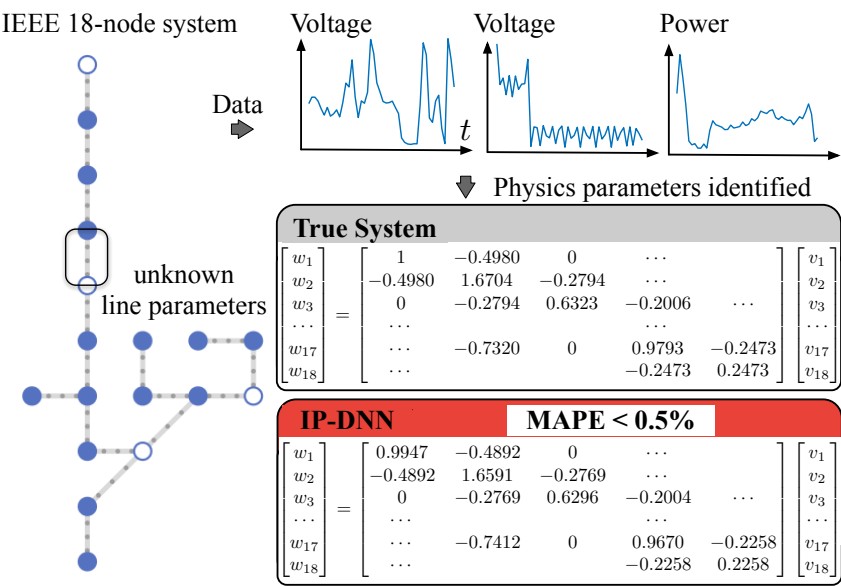

Figure 14: Line parameter identification in a partially-observable IEEE 18-node power grid system.

C.8 HEAT SOURCE IDENTIFICATION IN INVERSE HEAT TRANSFER PROBLEM

In this section, we describe our experiments conducted on a simulated heat transfer dataset to evaluate the performance of the proposed ODNN method for identifying true heat sources in a noisy 2D temperature distribution. We consider two heat source cases in our experiments. **Case f.** It represents the true heat transfer system, generated by four heat sources located symmetrically along the edges of the domain: the left center, upper center, right center, and bottom center. The resulting temperature distribution from Case f is *symmetric about both axes*, forming a consistent and balanced heat map. The temperature at any point $(x, y, t)$ in Case f can be represented by:

$$T_f(x, y, t) = \sum_{i=1}^{4} \frac{Q_i}{4\pi\alpha t} \exp\left(-\frac{(x - x_i)^2 + (y - y_i)^2}{4\alpha t}\right)$$

where $Q_i, i = 1, 2, 3, 4$ represents the strength of each heat source, and $(x_i, y_i)$ are the coordinates of the four symmetric sources:

- $(x_1, y_1) = (0, H/2)$: Located at the *left center edge* of the domain.
- $(x_2, y_2) = (L/2, H)$: Positioned at the *upper center edge* of the domain.
- $(x_3, y_3) = (L, H/2)$: Located at the *right center edge* of the domain.
- $(x_4, y_4) = (L/2, 0)$: Positioned at the *bottom center edge* of the domain.

Here, $L$ and $H$ represent the *length* and *height* of the rectangular domain, respectively. The choice of these coordinates ensures that the heat sources are symmetrically positioned along the centers of the four edge lines, resulting in an inherently symmetric temperature distribution across the domain. $\alpha$ is the thermal diffusivity. The symmetry of Case f implies that the contributions from the four sources are balanced and reflected across both axes.

In contrast, **Case g** introduces an additional heat source located at the *upper-left corner* of the domain, which acts as a source of *noise*. This heat source produces an *asymmetric temperature distribution* that skews the overall heatmap, making the identification of the original four sources a challenging problem. The temperature contribution from the heat source at the upper-left corner in Case g can be described as:

$$T_g(x, y, t) = \frac{Q_5}{4\pi\alpha t} \exp\left(-\frac{(x - x_5)^2 + (y - y_5)^2}{4\alpha t}\right)$$

where $Q_5$ is the heat strength of the noise source, located at $(x_5, y_5) = (0, H)$. The observed temperature distribution, $T_{\text{obs}}(x, y, t)$, is the superposition of the contributions from both cases:

$$T_{\text{obs}}(x, y, t) = T_f(x, y, t) + T_g(x, y, t)$$

The goal of our experiment is to determine whether the proposed **ODNN** method can accurately disentangle the contributions of the true sources (Case f) from the noisy influence introduced by Case g, and subsequently reconstruct the 2D temperature heatmap. An essential aspect of this experiment is the observation that the *heat transfer equation for Case f* is inherently *symmetric*, while the heat transfer from Case g, being a single point source in the upper-left corner, is *asymmetric*. This difference creates an *implicit orthogonality* between the symmetric and asymmetric components of the temperature distribution, which our ODNN approach exploits to disentangle the physics-based temperature distribution from the noise. By leveraging this implicit orthogonality, ODNN is capable of identifying the structural symmetries in the observed data, allowing it to accurately separate the true heat distribution from the noise contribution.

Figure 15 presents the results of our experiment, showing the identified temperature distribution and the absolute error between the identified result and the ground truth. In the *left panel*, the identified temperature heatmap, produced by ODNN, clearly shows the original four heat sources without significant influence from the noisy upper-left corner heat source, effectively reconstructing the symmetric temperature distribution of Case f. In the *right panel*, we plot the **absolute error** between the identified temperature map and the ground truth:

$$E(x, y, t) = |T_{\text{identified}}(x, y, t) - T_f(x, y, t)|$$

showing minimal discrepancies across the domain, with slightly higher error values localized near the noise source. Notably, our approach achieves a MAPE of *less than 1%*, demonstrating its accuracy and effectiveness in recovering the true temperature map. The results indicate that ODNN successfully leverages the inherent symmetry properties of the physical system, allowing it to separate the true physics from noise, even in cases of significant noise interference. This ability to disentangle contributions from orthogonal components makes ODNN a powerful tool for identifying true sources in complex heat transfer systems, providing highly accurate temperature reconstructions essential for thermal analysis, environmental modeling, and related engineering applications.

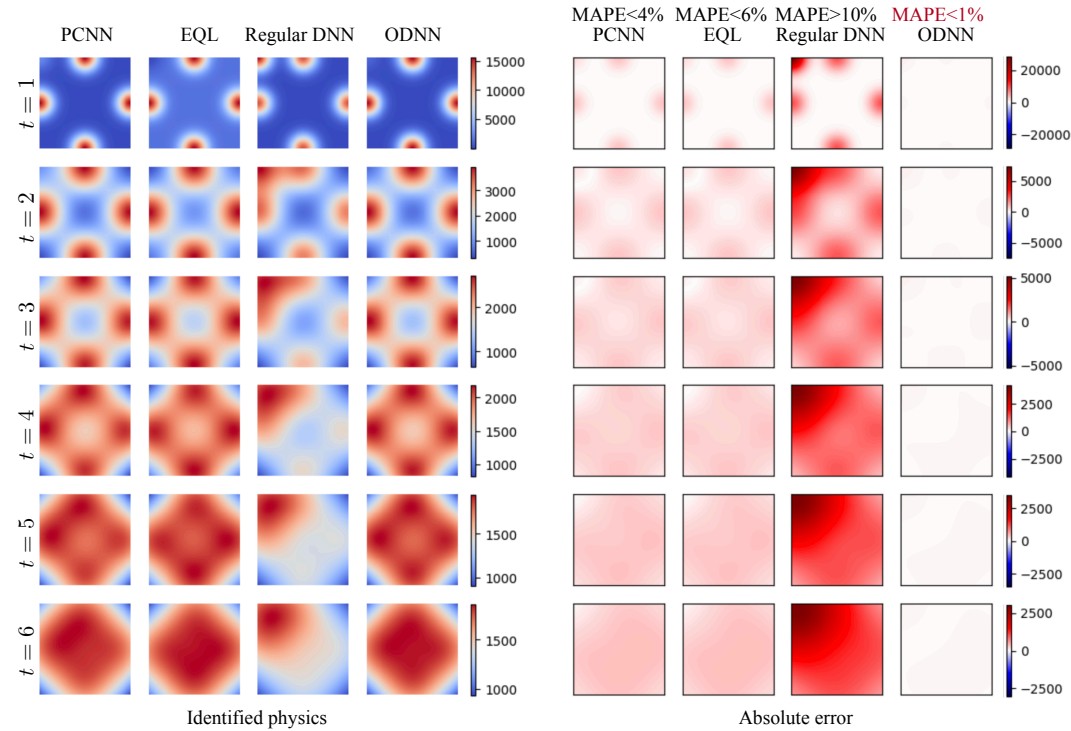

Figure 15: Temperature distribution identified in an inverse heat transfer problem.

## C.9 PHYSICS IDENTIFICATION IN BIOLOGY AND PENDULUM DATASETS

This subsection describes the experiment on Biology and Pendulum Datasets. **Driven pendulum dataset.** This system represents a classic mechanical system characterized by oscillatory motion. The movement $\theta(t)$ of driven pendulum can be expressed as $\theta(t) = \theta_0 \cos(\omega t) + \frac{f}{2\omega} t \sin(\omega t)$, where $\omega$ is the oscillator frequency and $f$ the unknown driven force. These two functions exhibit the "contrasting property" in terms of symmetry. We simulate this dataset using the underlying equations of pendulum motion. **Biology growth dataset.** This system describes bacterial population dynamics and represents a canonical population growth model. Following the Aiba-Edward model (Muloiwa et al., 2020), the population over time $\mu(t)$ can be modeled as $\mu(t) = \frac{S}{S+K_S}(\exp \frac{S}{K_S} + \cos(2\pi t))$ where $K_S$ is the half saturation constant and $S$ is the unknown substrate concentration. This model incorporates an underlying exponential growth component coupled with a sinusoidal disturbance simulating daily temperature fluctuations, thus satisfying the "contrasting property" requirement as periodicity. We simulate the time-series data using an Escherichia coli dataset following studies in (Paula et al., 2020; Aida et al., 2022).

In Figure 16, we present the identified physical equations for both the training and testing datasets, illustrating the superior generalization capability of the proposed ODNN compared to a regular DNN. This enhanced generalization is crucial for effective system control and robust decision-making in complex environments. Specifically, ODNN accurately identifies the physical dynamics even in the testing set, demonstrating its ability to learn and generalize the underlying physics beyond the training data. This highlights ODNN's success in preserving the integrity of the physical components while disentangling disturbances.

In contrast, regular DNNs exhibit inaccurate parameter estimation, especially in unseen data. Such inaccuracies have significant consequences: in biological systems, misestimating growth rates can lead to ineffective or harmful treatment plans, while in engineering, incorrect parameter values can compromise both system performance and safety. Thus, the ability of ODNN to accurately recover physical parameters, even in challenging scenarios, is essential for ensuring the reliability, safety, and effectiveness of real-world applications.

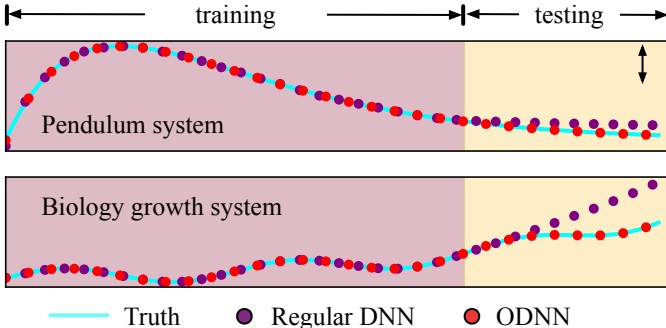

Figure 16: Accurate physics identification leads to better generalizability.

### C.10 EXTENDING TO MULTIPLICATIVE DYNAMICS: POPULATION GROWTH CASE STUDY

In addition to the additive setting $F(\mathbf{x}) = f(\mathbf{x}) + g(\mathbf{x})$, this section explores the multiplicative setting $F(\mathbf{x}) = f(\mathbf{x}) \cdot g(\mathbf{x})$ and demonstrates how the proposed ODNN method can adapt to such scenarios. We consider a synthetic time-series dataset modeling population growth as: $F(t) = (1+r)^t \cdot \left(1 + \frac{1}{2}\sin(k_0 t)\right)$ where $(1+r)^t$ represents the known physics basis function of exponential growth, $\left(1 + \frac{1}{2}\sin(k_0 t)\right)$ models seasonal effects on population growth with $k_0$ as the frequency parameter, and $t$ is the time. In this model, $r$ is the unknown growth rate we aim to learn. This synthetic case captures characteristics of many real-world population growth scenarios, such as those influenced by both exponential trends and periodic fluctuations.

To handle the multiplicative setting, we apply a logarithmic transformation to convert it into an additive form:

$$\bar{F}(t) = \log F(t) = t \cdot \log(1+r) + \log\left(1 + \frac{1}{2}\sin(k_0 t)\right).$$

The transformed data $\bar{F}(t)$ is then fed into the ODNN model, where:

- $f(t) = t$ represents the linear term associated with the growth rate,

- $g(t) = \log\left(1 + \frac{1}{2}\sin(k_0 t)\right)$ encodes the seasonal fluctuations and is distinct from $f(t)$ due to its periodic nature.

By leveraging the separability of $f(t)$ and $g(t)$ in terms of their periodic and non-periodic characteristics, ODNN is able to disentangle the contributions of exponential growth and seasonal effects, enabling accurate recovery of the unknown growth rate $r$. The learning trajectory of $r$ is illustrated in Figure 17. The results demonstrate that ODNN successfully converges to the true value of $r$, highlighting its capability to adapt to more complex scenarios beyond the additive setting. This showcases the flexibility and robustness of ODNN in handling multiplicative dynamics, effectively disentangling the underlying components even in challenging settings.

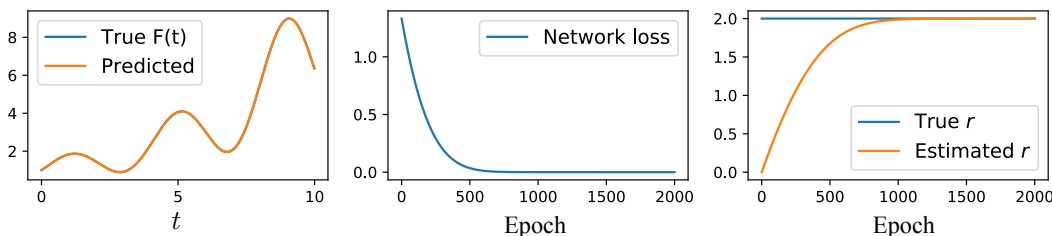

Figure 17: (Left) The predicted time-series data compared to true $F(t)$. (Middle) The ODNN loss function against epochs. (Right) The learning trajectory of the unknown parameter $r$.

## D LIMITATIONS AND FUTURE DIRECTIONS

While the ODNN framework demonstrates strong performance in disentangling physical components from disturbances and achieving accurate system identification, several potential limitations warrant further exploration. One key limitation is the reliance on an implicit or explicit assumption of orthogonality between physical dynamics and disturbance components. In complex systems where these conditions are not well-defined or where orthogonality is not easily identifiable, the performance of ODNN may be constrained. Additionally, the need for careful architectural design to ensure orthogonality might limit scalability when extending ODNN to more diverse or higher-dimensional systems. Another consideration is the computational cost associated with training constrained DNN architectures, which may be higher compared to unconstrained models, particularly for large-scale problems. Future research could focus on relaxing the orthogonality requirements, making the method applicable to a broader class of systems, as well as improving computational efficiency through more advanced optimization techniques. Exploring adaptive mechanisms to automatically identify suitable constraints could further enhance the applicability and robustness of ODNN across different real-world domains.

