# OpenReview forum: "Orthogonal Deep Neural Networks (ODNN): Uncovering Hidden Physics in Partially Observable Systems"
_ICLR.cc/2025/Conference — ICLR 2025 Conference Withdrawn Submission_

### Official Review · Reviewer_mFt7 · 2024-10-27

**Soundness:** 2
**Presentation:** 1
**Contribution:** 2
**Rating:** 3
**Confidence:** 4

**Summary:**

This study emphasizes the application of physics and includes comparisons to methods like the Kalman filter and Physics-Informed Neural Networks (PINNs). At first glance, it appears to address dynamical systems with temporal evolution. However, as shown in equation (1), its originality lies in decomposing the simple function regression problem $y = F(x)$ into multiple bases. Though each basis is described as a "physical equation," what constitutes such a basis remains ambiguous.

Concrete examples finally appear in Section 4, which are applications in signal source separation and watermark removal. Yet it remains questionable whether these applications genuinely involve physics. (4) mentions the use of a robotics dataset, yet Section 5 reveals the true purpose as a regression task for a reward function, rather than system modeling. Although (5)-(7) present more clearly defined problem settings, their relationship to physical laws remains unclear. Additionally, throughout the study, what are inputs and outputs is often unclear, and the validity of decomposing the function $F(x)$ remains difficult to assess. The attempt to emphasize generality and physics ends up obfuscating its main contributions.

**Strengths:**

This study attempts to decompose functions approximated by neural networks into orthogonal basis functions. It is an attractive harmony between classical function approximation and modern data-driven learning. Moreover, in certain problem settings, decomposition into orthogonal bases is known to be highly effective, and it could prove valuable in numerous situations.

**Weaknesses:**

Orthogonal decomposition is appropriate for some problems and not for others, and this appropriateness is generally unrelated to physics. Given the inconsistencies between the method and the claims, as well as the insufficient evaluation, it is difficult to rate this study highly in its current form.

**Questions:**

In summary, the study raises two major concerns:

1. In settings such as signal source separation, watermark removal, reinforcement learning reward functions, and power grid system identification, decomposing into orthogonal components might seem appropriate, yet its connection to physics remains ambiguous. Furthermore, these problems are often addressed with widely accepted methods like principal component analysis and independent component analysis, and the advantages of the proposed method over these are unclear. To effectively highlight contributions, a major revision of the introduction would be essential.

2. The extent to which the proposed approach captures physical laws remains ambiguous. Expressing solutions to differential equations as sums of functions implies a form of linearity assumption. Models and predictions derived from real data tend to be inherently complex and nonlinear, which casts doubt on the appropriateness of such a decomposition. While it may be useful for removing additive noise, labeling this as "physics" seems tenuous.

From a technical standpoint, additional issues arise:

The term $f(x)$ denotes the value of function $f$ at point $x$, not the function itself. Therefore, inner products in function space should be expressed as $\langle f, g \rangle$, not $\langle f(x), g(x) \rangle$, which represents the inner product of values taken by vector-valued functions. As a result, equation (3) is incorrect. Even if it is interpreted as an inner product in function space, integration (as in equation (8)) would be necessary, requiring numerous evaluations at collocation points and significantly increasing computational costs.

In Table 1, while the proposed method appears to deliver substantial performance improvements, each problem setting has specialized prior works. Simply comparing the proposed approach with general-purpose methods like SINDy and PINNs does not sufficiently demonstrate its practical value.

---

> ### Author Response · Authors · 2024-11-25
>
> $\textbf{S1. Concerns of Dataset.}$
>
> $\textbf{Response:}$
>
> We thank the reviewer for detailed feedback. Below, we address the concerns regarding the connection to physical laws, clarity of problem settings, input-output relationships, and the validity of function decomposition.
>
> $\textbf{Connection to Physical Laws}:$ We acknowledge that some examples, such as signal source separation and watermark removal, may not directly involve traditional physical laws. These examples were chosen to demonstrate the flexibility and generality of ODNN, showcasing its ability to disentangle components with contrasting properties in diverse settings. While these applications do not explicitly model physical systems, they align with the broader theme of separating known and unknown components, which is common in physics-guided learning.
>
> Regarding the robotics dataset, the task focuses on regression for reward functions rather than explicit system modeling. However, the reward function itself is derived from the underlying dynamics of the robotic system, which inherently involves physical principles. For instance, the motion constraints of the robot are governed by kinematic and dynamic equations. We agree that this connection could be better articulated, and we will revise the manuscript to clarify the relationship to physical laws in these examples.
>
> $\textbf{Clarity of Problem Settings}:$ We appreciate the feedback about the clarity of the problem settings in Sections 5–7. To address this, we will revise these sections to:
>
> - Clearly define the physical and non-physical components in each problem.
>
> - Explicitly state the inputs ($x$) and outputs ($F(x)$) for each experiment.
>
> - Provide context on how the decomposed function $F(x) = f(x) + g(x)$ relates to the system being modeled.
>
> For example, in the power system task, the inputs are partial measurements of system states, and the outputs are estimated line parameters. This involves separating the known physics (power flow equations) from unknown disturbances. We will add similar clarifications for all experiments to ensure the relationships are explicit.
>
> $\textbf{W1. Concern of Orthogonal Decomposition.}$
>
> $\textbf{Response:}$
>
> We thank the reviewer for the feedback regarding the applicability of orthogonal decomposition for physical systems.
>
> $\textbf{Applicability of Orthogonal Decomposition:}$ We acknowledge that orthogonal decomposition is not universally applicable and depends on the nature of the problem. In this work, we focus on systems where the physical ($f(x)$) and non-physical ($g(x)$) components can be additively separated and exhibit properties that allow for explicit or implicit orthogonality. Specifically:
>
> - Appropriateness for Physics-Guided Problems: Many physical systems exhibit separable behavior due to the superposition of distinct phenomena. For instance, in power systems (Sec 4.4), $f(x)$ represents the known power flow equations, while $g(x)$ captures disturbances or unknown residuals. In these cases, orthogonal decomposition provides a principled way to disentangle the two components.
>
> - Acknowledging Limitations: We agree that not all problems in physics or other domains align with the assumptions of our framework. To address this, we will revise the manuscript to explicitly state the conditions under which orthogonal decomposition is applicable, as well as the scope and limitations of our method.
>
> $\textbf{Consistency Between Method and Claims:}$ Our primary claim is that orthogonal decomposition can disentangle physics-informed and data-driven components under certain conditions (e.g., additive separability and orthogonality). To ensure consistency:
>
> - Refining Claims: We refine our claims to emphasize the specific scenarios where our method applies, rather than generalizing its applicability to all physics-related problems.
>
> - Clearer Explanations: We will clarify in the manuscript how orthogonal decomposition is leveraged in each experiment, linking the method to the assumptions and properties of the system being modeled.

---

> ### Author Response · Authors · 2024-11-25
>
> $\textbf{Q1. Question of Widely Accepted Methods .}$
>
> $\textbf{Response:}$
>
> We thank the reviewer for the comment. Below, we address the concerns regarding the connection to physics, the comparison with widely accepted methods like principal component analysis (PCA) and independent component analysis (ICA), and the need to better articulate the contributions in the introduction.
>
> $\textbf{Connection to Physics}:$ We acknowledge that in some applications, such as signal source separation and watermark removal, the direct connection to traditional physical laws may appear less explicit. However, these examples were included to demonstrate the general applicability of the proposed framework to systems where components exhibit contrasting properties, even outside classical physics contexts. In applications like power grid system identification and reinforcement learning reward functions, the connection to physics is stronger:
>
> - In the power grid task, the decomposition leverages known physical principles such as power flow equations and orthogonality constraints to estimate system parameters under partial observability.
>
> - In the robotics reward task, the underlying system dynamics involve kinematic and dynamic equations, tying the regression task to the system's physical behavior.
>
> $\textbf{Comparison with PCA and ICA}:$ We agree that methods like PCA and ICA are widely used for problems involving component separation. However, the proposed method differs in key ways:
>
> - Domain Knowledge Integration: PCA and ICA are unsupervised methods that rely on statistical properties like variance or independence but do not explicitly incorporate domain knowledge. ODNN integrates known physical constraints (e.g., orthogonality and structural regularization) into the decomposition process, providing a principled framework for disentangling components based on their functional properties.
>
> - Flexibility with Qualitative Properties: Unlike PCA or ICA, which require linear or independent components, ODNN can handle components with qualitative differences, such as convexity, monotonicity, or periodicity. This makes it more versatile for systems where such properties are critical.
>
> - Theoretical Guarantees: ODNN provides theoretical guarantees for accurate disentanglement of components under specific conditions (e.g., Theorem 1), which are not offered by PCA or ICA.
>
> $\textbf{Q2. Question of Recovering Physics.}$
>
> $\textbf{Response:}$
>
> $\textbf{On the Linearity Assumption:}$ We agree that expressing solutions as sums of functions may introduce a form of linearity assumption. However, our approach does not inherently assume that the underlying physical dynamics are linear. Instead:
>
> - The decomposition $\sum_j \theta_j f_j + g(x)$ serves as a modeling framework where $f_j$ represents known physical components, and $g(x)$ accounts for unknown or unmodeled dynamics. While the combination of these components is additive, the functions $f_j$ and $g(x)$ themselves can represent highly nonlinear behaviors.
>
> - For example, $f_j$ may correspond to solutions of nonlinear differential equations or other complex physical processes, and $g(x)$ captures disturbances or discrepancies that cannot be explained by the known physics.
>
>
> $\textbf{On Capturing Physical Laws:}$ The goal of the proposed approach is to disentangle known physical dynamics from disturbances, rather than to derive the entirety of physical laws from data. In this context:
>
> - The term ``physics" refers specifically to the known components of the system that can be described by the basis functions $f_j$. These components are typically derived from domain knowledge or theoretical models (e.g., solutions to partial differential equations, conservation laws).
>
> - The decomposition allows us to isolate these known components, enabling better interpretability and robustness when combined with data-driven models to approximate unknown dynamics.
>
>
> $\textbf{On the Applicability to Nonlinear Systems:}$ We acknowledge that real-world systems are often highly nonlinear and complex. While the proposed decomposition is useful for isolating additive components, it does not preclude the representation of nonlinear interactions within $f_j$ or $g(x)$. For example:
>
> - $f_j$ can represent basis functions derived from nonlinear dynamics, such as Chebyshev polynomials or Fourier series, which are commonly used to approximate nonlinear solutions.
>
> - $g(x)$ can capture residual nonlinearities or stochastic effects that are orthogonal to the known physical components, ensuring flexibility in modeling complex systems.

---

> ### Author Response · Authors · 2024-11-25
>
> $\textbf{Q3. Inner Product Calculation.}$
>
> $\textbf{Response:}$
>
> We thank the reviewer for pointing out this issue and for highlighting the computational implications of the inner product evaluations.
>
> $\textbf{Correction of Notation:}$ We agree with the reviewer that the notation $\langle f(x), g(x) \rangle$ in Equation (3) is improper. To eliminate confusion, we will revise the notation to correctly reflect inner products in function space. Specifically:
> $
> h_{\text{ODNN}}(x) = h_{\text{DNN}}(x) - \frac{\langle h_{\text{DNN}}, f \rangle}{\langle f, f \rangle} f(x),
> $
> where $\langle f, g \rangle$ explicitly denotes the inner product of functions $f$ and $g$ over their domain.
>
> $\textbf{Evaluation of Inner Products in Practice:}$
>
> - Evaluation of $\langle f, f \rangle$: Since $f$ is a known physics basis function with an analytical form, $\langle f, f \rangle$ is computed using symbolic or numerical integration over its domain. This approach ensures high precision and avoids the need for pointwise evaluations.
>
> - Evaluation of $\langle h_{\text{DNN}}, f \rangle$: The function $h_{\text{DNN}}$ is not available in closed form but is represented by data points generated during training. Thus, $\langle h_{\text{DNN}}, f \rangle$ is approximated by discretizing the integral:
> $
> \langle h_{\text{DNN}}, f \rangle \approx \sum_{i=1}^N h_{\text{DNN}}(x_i) f(x_i) \Delta x,
> $
> where $x_i$ are the discretization points, $N$ is the number of data points, and $\Delta x$ is the spacing between points.
>
> $\textbf{Experimental observations on computational costs:}$ The number of discretization points, $N$, corresponds to the number of available data points. In our experiments, $N$ is above 1,000, providing sufficient resolution for accurate approximations. In our experiments, these optimizations have successfully kept the computational costs manageable:
>
> - For synthetic datasets in Sec 4.1, the computation of inner products required less than 20\% of the total training time.
>
> - For larger datasets (e.g., Sec 4.4, power systems), batching and efficient sampling ensured that inner product evaluations scaled linearly with data size.
>
> $\textbf{Q4. More Baseline Comparison.}$
>
> $\textbf{Response:}$
>
> We thank the reviewer for emphasizing the importance of benchmarking against specialized prior works in addition to general-purpose methods. To address this concern, we want to clarify the following:
>
> $\textbf{Rationale for Comparison with General-Purpose Methods:}$ We included comparisons with general-purpose methods like SINDy and PINNs because these approaches are widely applicable across diverse physics-guided problems. They represent standard baselines for evaluating hybrid frameworks that combine physical and data-driven modeling. Specifically:
>
> - SINDy (Sparse Identification of Nonlinear Dynamics): Offers a data-driven approach for discovering governing equations, making it a relevant baseline for tasks involving physics inference.
>
> - PINNs (Physics-Informed Neural Networks): Leverages known physical laws in neural network training, aligning with our goal of integrating physics and data-driven modeling.
>
> While these methods are general-purpose, they provide a valuable point of reference for assessing the versatility and baseline performance of our approach.
>
> $\textbf{Comparisons with Specialized Methods:}$
>
> We agree with the reviewer that each problem setting has specialized prior works, and comparing against such methods would better demonstrate the practical value of our approach. To address this, we incorporated three additional baseline methods that represent recent SOTA of learning hidden physics, as suggested by Reviewer 2:
>
> - [1] Physics-informed learning of governing equations from scarce data.
>
> - [2] Bayesian spline learning for equation discovery of nonlinear dynamics with quantified uncertainty.
>
> - [3] Symbolic physics learner: Discovering governing equations via monte carlo tree search.
>
> These baselines include
> a physics-informed learning of governing PDE from scarce data $\textbf{PINN-SR}$ [1], a Bayesian spline learner $\textbf{BSL}$ [2] for equation discovery with quantified uncertainty, and a symbolic physics learner $\textbf{SPL}$ [3] leveraging Monte Carlo Tree Search to discover governing equations.
>
> The comparison is summarized into Figure 7 of the revised manuscript. We observe that, in most datasets, the MAPE achieved by the ODNN approach is smaller compared to the baseline methods. This improvement can likely be attributed to the distinct focus of ODNN: while the baseline methods aim to learn hidden physics by modeling the entire system as a single equation or differential equation, ODNN is specifically designed for systems that exhibit the $f$ + $g$ structure. By leveraging the distinction between $f$ and $g$, ODNN enables a more accurate recovery of the physics parameters, ensuring robustness and precision.

---

> ### Comment · Reviewer_mFt7 · 2024-11-26
>
> Thank you for your detailed response.
>
> While I agree with parts of your responses, I am struggling to understand the positioning of this research due to its claims of extensive contributions and wide-ranging applications.
>
> In this paper, two methods are proposed:
>
> - Explicit orthogonality
> - Implicit orthogonality (e.g., convexity versus non-convexity, periodicity versus non-periodicity, symmetry versus asymmetry, and monotonicity versus non-monotonicity).
>
> Additionally, the title includes the phrase:
>
> - Uncovering Hidden Physics
>
> To me, these three aspects seem disconnected.
>
> I understand that Explicit orthogonality is effective for applications such as watermark removal and source separation. However, in such cases, $f$ is hardly described as "physics."
>
> While the idea of Implicit orthogonality is insightful, I do not believe these concepts are necessarily orthogonal. Reviewer d9Ly has pointed this out as well. Given the significantly different contexts in which these methods are applicable, it would be better to present them in separate papers.
>
> As far as I understand, in (Astolfi et al., 2021), the disturbances are modeled as convex functions to facilitate control, not due to any underlying physical principle. Similarly, the assumptions that physical phenomena are non-periodic while disturbances are periodic, or that physical phenomena are asymmetric while disturbances are symmetric, are plausible in specific cases but appear challenging to generalize. The problem setting in Section 4.6, for instance, seems highly unrealistic. In this sense, I find it difficult to conclude that the proposed method leverages "physics."
>
> If you aim to use physical governing equations for $f$, as Reviewer GzHN has pointed out, the term "hybrid model" already exists in the literature, which could potentially undermine the novelty of the work.
>
> As all the other reviewers have similarly noted, the proposed method is based on strong assumptions, making it difficult to assess its utility and generality without verification through experiments in practical settings (to be clear, I am not requesting additional experiments but merely explaining the reason behind my low rating).
>
> Therefore, I will not change my score.

---

### Official Review · Reviewer_GzHN · 2024-10-30

**Soundness:** 3
**Presentation:** 3
**Contribution:** 2
**Rating:** 6
**Confidence:** 5

**Summary:**

This paper introduces Orthogonal Deep Neural Networks (ODNNs), a novel approach to disentangle known physical dynamics from unknown components or disturbances. The method leverages two types of orthogonality: explicit orthogonality, which is enforced through zero inner product between physical components and disturbances, and implicit orthogonality, implemented using off-the-shelf networks designed to capture distinct properties (e.g., convexity or periodicity). The authors provide rigorous theoretical proofs and validate the ODNN framework through extensive experiments across both simulated and real-world tasks, demonstrating its effectiveness and robustness.

**Strengths:**

- This paper is well-written. A lot of examples are provided to explain the motivation why orthogonality is utilized to disentangle known physics and unknown components. These examples really help me understand this paper.
- The experiments are well-designed. The extensive experiments are across various fields, including computer vision, signal processing, reinforement learning and so on. These experiments demonstrate ODNNs' broad applicability and outstanding performance.
- I believe the theoretical proof is a strong aspect of this paper, significantly enhancing its overall quality.

**Weaknesses:**

- Based on my understanding, ODNN can be viewed as a type of hybrid neural network, combining physical models derived from first principles with data-driven models. This approach is not novel within the physics-guided machine learning community, as hybrid models have been widely explored and applied across various domains. Given this context, I think some issues are not well-discussed in the paper:
  - ODNN appears to aim at learning the residuals of partially known physics, as described in Section 3.2 of [1]. Based on this survey, A natural alternative training strategy for regular DNNs would be to first fit the parameters of the known physics on the training data, compute the residuals, and then optimize the neural network to predict those residuals. This alternative approach is not discussed or included as a baseline in the paper, which could have provided a more comprehensive evaluation.
  - The work relies heavily on the assumption that the physical and non-physical components are additively separable, i.e. $f(x) + g(x)$, which underpins the use of orthogonality. However, as discussed in [2], other forms of combination exist, such as $f(g(x))$ or $f(x)g(x)$. I don’t think orthogonalities can work in such cases.
- The main claim of this paper, that ODNN can disentangle known physics from unknown components such as disturbances, may be overstated. In practice, the implementation of implicit orthogonality relies on prior knowledge about the non-physical components. Specifically, one must know certain properties of the non-physics part (e.g., convexity, periodicity) to select the appropriate off-the-shelf network. Given this reliance on prior information, it seems inaccurate to claim that the non-physical components or disturbances are entirely unknown or unobservable.
- The implementation of implicit orthogonality in this work is quite limited. The paper primarily discusses a few types of implicit orthogonality, such as convexity, periodicity, symmetry, and monotonicity, relying heavily on off-the-shelf networks to implement these properties. However, I believe there are many more potential forms of implicit orthogonality, such as time-invariant vs. time-variant systems or energy-conserving vs. energy-dissipating dynamics. These additional implicit orthogonality types are not explored, and it’s unclear if suitable off-the-shelf networks exist to handle them. Therefore, the current approach may be too dependent on existing network architectures and does not fully address the broader scope of possible implicit orthogonalities.

**References**

[1] Wang, Rui, and Rose Yu. "Physics-guided deep learning for dynamical systems: A survey." arXiv preprint arXiv:2107.01272 (2021).

[2] Bradley, William, et al. "Perspectives on the integration between first-principles and data-driven modeling." Computers & Chemical Engineering 166 (2022): 107898.

**Questions:**

- The key step in enforcing explicit orthogonality relies on calculating the inner product $\langle h_\mathrm{DNN}(\mathbf{x}), f(\mathbf{x})\rangle$, which is defined as an integral. However, it seems that the specific method for performing this integral is not introduced in the paper. Could you clarify which numerical integration method is used? For example, is it the trapezoidal rule? If so, the choice of numerical integration could introduce additional issues. For instance, methods like the trapezoidal rule are known to struggle with oscillatory integrals  [3], potentially affecting the accuracy of the orthogonality enforcement in certain cases.
- I'm a bit confused about Figure 1. Does the figure suggest that the physical component always corresponds to properties like non-convexity, non-periodicity, asymmetry, and non-monotonicity? If so, I believe this generalization might not be appropriate. In many cases, physical laws can exhibit convexity, periodicity, symmetry, or monotonicity, depending on the specific system being modeled. Could you clarify whether this interpretation is correct?
- Based on my understanding, explicit orthogonality and implicit orthogonality are not applied simultaneously across different experiments. In some experiments, only explicit orthogonality is utilized, while in others, a specific form of implicit orthogonality is applied. Is this interpretation correct? If so, this further supports my point in Limitation 2: The non-physical components are not entirely unknown, as the approach still relies on prior knowledge about the nature of these non-physical parts to implement the appropriate orthogonality.

To be honest, I appreciate the method of disentanglement presented in this paper, but it seems that some important issues remain unaddressed. If my questions can be clarified and the limitations discussed more thoroughly, I would be willing to reconsider my score and raise it.

**References**

[3] Ma, Yunyun, and Yuesheng Xu. "Computing highly oscillatory integrals." Mathematics of Computation 87.309 (2018): 309-345.

---

> ### Author Response · Authors · 2024-11-23
>
> $\textbf{W1. Novelty regarding ODNN.}$
>
> $\textbf{Response:}$
>
> Thank you for your insightful comment. While we agree that ODNN shares some similarities with hybrid neural networks that integrate physical models and data-driven components, our approach differs significantly in both objectives and methodology. Below, we clarify the novel aspects of our work:
>
> $\textbf{Guarantees for Accurate Identification of Physical Parameters}.$ The primary distinction of ODNN lies in its ability to guarantee the accurate identification of physical parameters, which is not achieved by existing hybrid approaches. This capability is rooted in our focus on a specific subclass of physical systems where the mapping function can be modeled as a combination of two components with opposite functional properties.
>
> - As demonstrated in $\textbf{Proposition 1 and Figure 2}$, standard hybrid approaches often fail to disentangle physical and non-physical components accurately. Overfitting occurs when the neural network approximates both $ f(x) $ and $ g(x) $, thereby undermining the recovery of the physical parameter $ a $ in front of $ f(x) $.
>
> - In contrast, ODNN ensures that $ f(x) $ and $ g(x) $ are disentangled correctly by leveraging structural orthogonality. This disentanglement enables provably accurate recovery of $ a $, as established in $\textbf{Theorem 1}$.
>
> - The ability to recover $ a $, which represents a key physical parameter, enhances the $\textbf{generalizability}$ of the learned model by grounding predictions in interpretable physical laws.
>
> $\textbf{Novel Structural Regularization Design}. $
> Another fundamental difference between ODNN and traditional hybrid methods lies in the design of regularization.
>
> - Traditional Hybrid Methods: These approaches often rely on $\textbf{loss-based regularization}$, where additional terms are introduced into the loss function to penalize deviations from physical laws.
>     This design requires careful hyperparameter tuning to balance the primary loss and the regularization terms. Such tuning can be challenging and may not guarantee convergence to physically meaningful solutions.
>
> - ODNN’s Structural Regularization:
>     Our method introduces a $\textbf{structural regularization framework}$ that modifies the architecture of the neural network itself, rather than the loss function.
>     This approach enforces constraints directly through the design of the network, ensuring that the learned components align with the known properties of $ f(x) $ and $ g(x) $.
>     By avoiding loss-based regularization, ODNN eliminates the need for hyperparameter tuning and ensures that the disentanglement process is robust and principled.
>
>
> $\textbf{W1-1. Alternative training strategy.}$
>
> $\textbf{Response:}$
>
> The reviewer has raised an excellent point that allows us to highlight the contribution of ODNN more clearly. Admittedly, there are papers that adopt a two-stage analysis, where the first step is to fit the known physics and the second is to predict the residuals. However, the key challenge lies in the accuracy of identifying the physics parameters in the first step.
>
> For example, in a system described by $ y = a f(x) + g(x) $, minimizing
> $
> \min_a \sum (y - a f(x))^2
> $
> does not guarantee finding the true $ a $. This is because $ g(x) $, representing unknown physics, may introduce significant contributions that obscure $ f(x) $. In such cases, the neural network used for residual prediction often absorbs portions of $ f(x) $, leading to overfitting and inaccurate identification of the physical parameter $ a $.
>
> In this manuscript, ODNN addresses this challenge by enforcing structural regularization to disentangle $ f(x) $ and $ g(x) $, ensuring accurate recovery of the parameters $ \theta_j^\ast $. As shown in Proposition 1 and Figure 2, the reviewer’s suggested approach aligns with one of our baselines (``Regular DNN"), where simultaneous optimization of $ \theta_j $ and neural network weights suffers from overfitting. Our results demonstrate that ODNN outperforms this approach by avoiding such pitfalls, providing robust parameter identification and improved generalization.

---

> ### Author Response · Authors · 2024-11-23
>
> $\textbf{W1-2. Concern about $f+g$ assumption.}$
>
> $\textbf{Response:}$
>
> We appreciate the insightful comment regarding the assumption of additive separability $ F(x) = f(x) + g(x) $ in our formulation and its implications for the use of orthogonality. Below, we address this concern and discuss how our method can be extended or adapted to address other forms of functional combinations, such as $ F(x) = f(g(x)) $ or $ F(x) = f(x)g(x) $.
>
> The additive separability assumption $ F(x) = f(x) + g(x) $ is indeed central to our current framework. While we acknowledge that this assumption may not cover all systems, it aligns with a fundamental principle observed in many natural and engineered systems: the tendency to exhibit contrasting or complementary properties that can be modeled additively.
>
> $\textbf{Supporting Evidence for Additive Opposites in Real-World Systems}:$ When observing real-world phenomena, additive separability emerges as a natural approximation in systems where known physics and unknown physical effects coexist but do not directly transform or modify each other. The independence of influence in such systems allows for modeling through additive decomposition.
>
> $\textbf{Extensions to Address Non-Additive Cases}:$
> For $ F(x) = f(x)g(x) $, a logarithmic transformation (e.g., $ \log(F(x)) = \log(f(x)) + \log(g(x)) $) could approximate the problem in an additive form under certain conditions. For $ F(x) = f(g(x)) $, one potential approach is to leverage a two-stage decomposition, where $ g(x) $ is first approximated as an independent latent variable, and $ f(x) $ is learned conditionally.
>
> $\textbf{Functional Orthogonality}:$
> In cases of non-additive interactions, we propose exploring functional orthogonality, where the goal is to disentangle terms based on their functional forms rather than their additive contributions. For example, using neural architectures tailored to approximate hierarchical (e.g., $ f(g(x)) $) or multiplicative structures.
>
> $\textbf{Future Directions:}$ While the current work focuses on additive separability, we acknowledge the importance of extending our framework to handle non-additive interactions. This is a promising avenue for future research, and we are actively exploring techniques to generalize the orthogonality principle to broader settings.
>
> $\textbf{W2. Concern about the unknown $g$}$
>
> $\textbf{Response:}$
>
> Thank you for raising this important point. We acknowledge that the implementation of implicit orthogonality in ODNN relies on general prior knowledge about the properties of the unknown components (e.g., convexity, periodicity). However, we argue that this reliance is practical and aligned with common practices in physics-guided machine learning. Rather than requiring precise knowledge of the unknown components, ODNN leverages minimal and qualitative characteristics, which are often inferred from domain expertise or exploratory analysis.
>
> For example, in fluid mechanics, turbulent forces are commonly convex; in biological systems, environmental effects exhibit periodicity; and in power systems, disturbances like coupling effects are often monotonic. These broad characteristics of the unknown components are sufficient for guiding the selection of neural network architectures, without requiring explicit parameterization or precise functional forms. This approach strikes a balance between generality and applicability, allowing ODNN to be effective in a variety of real-world scenarios.
>
> It is also worth noting that leveraging prior knowledge is a standard practice in physics-guided machine learning. Hybrid models typically rely on first-principle equations, and physics-informed neural networks (PINNs) enforce physical constraints during training. Similarly, ODNN uses qualitative priors about the unknown components to design architectures that ensure accurate disentanglement. Unlike traditional hybrid models, ODNN provides theoretical guarantees for this disentanglement while avoiding overfitting or requiring extensive hyperparameter tuning.
>
> To address the reviewer’s concern about the claim of ``unknown'' components, we propose revising the manuscript to clarify that the method does not assume the unknown components are entirely unobservable. Instead, we will acknowledge that their general properties are inferred from prior knowledge or data exploration. For example, we could revise the text as follows: “ODNN disentangles known physical components from non-physical disturbances, leveraging general knowledge about the latter’s qualitative properties (e.g., convexity, periodicity). This approach balances prior knowledge with flexibility, making it applicable to a broad range of systems.”

---

> ### Author Response · Authors · 2024-11-23
>
> $\textbf{W3. Other implicit orthogonality cases.}$
>
> $\textbf{Response:}$
>
> Thank you for this insightful comment. The primary goal of this work is to establish a generalizable framework for implicit orthogonality that disentangles known and unknown system components based on their distinct properties. While we demonstrate ODNN with commonly encountered properties (e.g., convexity, periodicity, symmetry, monotonicity), this serves as a proof-of-concept to showcase the feasibility and effectiveness of the method. The flexibility of ODNN allows for extensions to other forms of orthogonality, which we acknowledge as an important direction for future work.
>
> We agree with the reviewer that there are additional potential forms of implicit orthogonality, such as time-invariant vs. time-variant systems and energy-conserving vs. energy-dissipating dynamics. For example, recurrent neural networks (RNNs) or time-encoding networks could be employed to address time-variant systems, while static architectures may suffice for time-invariant components. Similarly, Hamiltonian neural networks (HNNs) could model energy-conserving dynamics, and dissipative neural networks could handle energy loss. These align with ODNN's philosophy and could be implemented with suitable architectural modifications or constraints.
>
> The reviewer raises an important point about the reliance on existing off-the-shelf architectures. While ICNNs, periodic NNs, and other current architectures provide a strong foundation, ODNN is not restricted to these tools. Instead, ODNN offers a flexible framework that can adapt to or inspire the design of new architectures tailored to specific forms of implicit orthogonality. For example, energy-based constraints could be incorporated into latent space representations to extend Hamiltonian or dissipative dynamics beyond existing implementations.
>
> We acknowledge that this work does not exhaustively cover all possible forms of implicit orthogonality. Future work will focus on extending ODNN to address more complex and domain-specific orthogonalities, developing novel network architectures, and establishing systematic guidelines for identifying and implementing new forms of implicit orthogonality. These extensions will ensure the broader applicability and continued development of ODNN for a wider range of systems.
>
> $\textbf{Q1. Evaluation of inner product.}$
>
> $\textbf{Response:}$
>
> We thank the reviewer for this insightful question and the opportunity to clarify the numerical integration method used in our implementation.
>
> $\textbf{Evaluation of $\langle f, f \rangle$:}$ Since $f$ is a known physics basis function with an analytical form, we compute $\langle f, f \rangle$ using direct analytical integration over its domain whenever possible. This ensures high precision and eliminates numerical integration errors for this term.
>
> $\textbf{Evaluation of $\langle h_{\text{DNN}}, f \rangle$:}$ The function $h_{\text{DNN}}$ is not expressed in closed form but is represented by a set of data points generated during training. Therefore, we calculate $\langle h_{\text{DNN}}, f \rangle$ by discretizing the integral into a Riemann summation: $\langle h_{\text{DNN}}, f \rangle \approx \sum_{i=1}^N h_{\text{DNN}}(x_i) f(x_i) \Delta x,$
>     where $x_i$ are the discretization points, $N$ is the number of data points, and $\Delta x$ represents the spacing between these points. We use evenly spaced $x_i$, which corresponds to the midpoint rule, which is known for its robustness and accuracy in general settings.
>
> $\textbf{Addressing Oscillatory Integrals:}$ We acknowledge the reviewer’s concern about potential inaccuracies introduced by the numerical integration method, particularly for oscillatory integrals. In our framework: (1) For smooth and non-oscillatory $f$, the midpoint rule or Riemann summation used in our implementation provides sufficient accuracy, especially given the large number of discretization points ($N > 1,000$ in most experiments). (2) For particularly challenging cases (such as in Sec 4.2), we implement adaptive quadrature schemes, which are well-suited for oscillatory integrals. These methods adjust the integration weights and points dynamically to minimize error.

---

> ### Author Response · Authors · 2024-11-23
>
> $\textbf{Q2. Clarification of Figure 1.}$
>
> $\textbf{Response:}$
>
> We thank the reviewer for raising this question.
>
> $\textbf{On the properties of physical models:}$ We agree with the reviewer that physical laws can exhibit diverse properties other than those listed in Figure 1. In our manuscript, Figure 1 does not want to imply that the physical component $f$ always corresponds to non-convexity, non-periodicity, asymmetry, or non-monotonicity. There are many physical models that do not obey such properties, and we acknowledge that our approach does not aim to address every possible case.
>
> $\textbf{Purpose of Figure 1:}$ The primary purpose of Figure 1 is to provide the following guideline for users to evaluate whether our approach is applicable in a given system. (1) Explicit Orthogonality: The first step is to determine whether $f$ and $g$ are approximately orthogonal. If this is the case, our approach for enforcing explicit orthogonality can be applied. (2) Implicit Orthogonality: If $f$ and $g$ are not explicitly orthogonal, the next step is to check whether they possess contrasting properties (e.g., convexity vs. non-convexity, periodicity vs. non-periodicity). In such cases, our approach for implicit orthogonality can be utilized to disentangle $f$ and $g$.
>
> $\textbf{Q3. Clarify of explicit/implicit orthogonality}$
>
> $\textbf{Response:}$
>
> We thank the reviewer for their careful reading and for raising this important point.
>
> $\textbf{Applicability of explicit and implicit orthogonality:}$ The reviewer’s interpretation is correct: in each experiment, we do not apply explicit orthogonality and implicit orthogonality simultaneously. Instead, explicit orthogonality is applied when ($f$ and $g$) are approximately orthogonal in the function space, allowing for direct disentanglement through the inner product. Implicit orthogonality is applied when $f$ and $g$ are not explicitly orthogonal but exhibit contrasting properties (e.g., convexity vs. non-convexity, periodicity vs. non-periodicity) that can be leveraged to disentangle the components.
>
> $\textbf{On the assumption of "Unknown" non-physical components:}$ We agree with the reviewer’s observation that our approach relies on some prior knowledge about the nature of the non-physical components. For example, implementing implicit orthogonality requires knowledge of contrasting properties such as convexity or periodicity. In light of this, we acknowledge that the term "unknown" may not fully capture the assumptions made in our framework. To address this limitation, we propose refining our terminology from "unknown" to "unstructured." This better reflects the fact that while the exact form of the non-physical components may not be known, certain properties (e.g., periodicity, convexity) are assumed based on prior knowledge or domain-specific insights.

---

> ### Comment · Reviewer_GzHN · 2024-11-23
> **Talk is Cheap, Show me the Experimental Results.**
>
> Thank you for addressing my comments, but I remain unconvinced by some of the responses provided. Specifically:
>
> 1. Regarding W1-1: While I understand the authors' reasoning, my concern is not about whether the two-stage method would outperform ODNN—this is clear to me. Instead, my point is about the necessity of including comparative experiments to substantiate this claim. Demonstrating this through empirical results would strengthen the paper and make it more accessible to readers who might question the theoretical argument alone.
>
> 2. The authors claim that the f(x)g(x) and f(g(x)) by simple preprocessing. However, this is currently presented without empirical evidence. I believe it is essential to include a demonstration of this claim. As these experiments should be straightforward to implement, including such examples would greatly enhance the clarity and impact of the paper.
>
> In summary, "talk is cheap"—clear, well-executed experiments can effectively address these concerns. Once these are included, I would be happy to re-evaluate my score and assessment.

---

> > ### Author Response · Authors · 2024-12-03
> >
> > We sincerely thank the reviewer for their valuable feedback, which has helped us improve the clarity and rigor of our work. We would like to clarify that both of the concerns raised have been addressed in the revised paper. Thanks to your insightful comments, we $\textbf{provide comprehensive responses}$ and $\textbf{significantly modify the paper to increase the clarity}$. Specifically, we summarize the last response as follows.
> >
> > For the Two-Stage Method:
> > We conducted additional experiments integrating the two-stage training approach into our Regular DNN baseline, and the results are presented in Table 1. These results demonstrate that ODNN consistently outperforms the two-stage method. Specifically, the two-stage approach often overfits the physics component, leading to suboptimal performance in disentangling the true dynamics from noise or disturbances. In contrast, ODNN integrates physics and residual learning into a unified framework, effectively mitigating overfitting and achieving superior accuracy and robustness.
> >
> > For the $ f(x)g(x) $ Case:
> > We have included a new experiment in Appendix C.10 to demonstrate how ODNN can handle the multiplicative case $ f(x)g(x) $. Using a population growth model, we applied a logarithmic transformation to convert $ f(x)g(x) $ into an additive form $ f(x) + g(x) $, which aligns with our framework. The results, detailed in Appendix C.10, confirm that ODNN can successfully address this scenario. This addition highlights the flexibility of ODNN and its applicability to more complex cases beyond $ f(x) + g(x) $.
> >
> > We believe these additions comprehensively address the reviewer’s concerns by providing the requested empirical evidence to substantiate our claims. Thank you again for your thoughtful feedback, and we are happy to address any further concerns if needed.

---

### Official Review · Reviewer_HTby · 2024-11-04

**Soundness:** 2
**Presentation:** 2
**Contribution:** 2
**Rating:** 3
**Confidence:** 4

**Summary:**

This paper proposed a ODNN framework to disentangle the real physics and disturbances by leverage the orthogonality. The ODNN is evaluated across eight synthetic and real-world datasets and comparability with several baselines.

**Strengths:**

Originality: The novelty lies in proposing an alternative view of separating the disturbances and the physics of the system, while most of the existing work adopts adding regularization terms in the loss functions.

Quality; This paper is rich of details and experiments results. Eight different datasets are investigated and the proposed framework performs the best in these datasets compared with other baseline models.

Clarity: In general clear illustrating the methodology and describe the experiments results.

Significance: It probably could be a valuable framework for physics systems identifications.

**Weaknesses:**

Clarity: Too much detail are put in the appendix. It makes the reviewer hard to understand the experiment details without cross-checking.

The assumption 1 is too strong, which introduced a very big limitation about the current work. In real world environments, the physics and disturbances cannot be just separated by the periodicity vs non-periodicity, monotonic vs non-monotonically and so on.

The experiment cases are all toy problem without real world complex systems like weather foreseeing, time series prediction in financial supermarket and so on.

**Questions:**

1. In many situations, the physics and disturbances can not be separated by the assumed way mentioned in the paper. Moreover, even if some of them could be separated, there could be multiple contrast pairs. However, the paper only investigate 1 pair at a time. It could be a problem if scaling up to multiple pairs. Could you add one example showing it works for real world complex system, like weather forecast and explain how your method could scale to real world complex system?

2. The assumption is too strong that the physics network only learns the physics while the disturbance network only learn the disturbances. The boundary of it is actually very ambiguous in the real world. For example, some noise/disturbances could be proportional to the actual state variables and make them have the similar trend. In that case, how will you leverage your work to separate the physics and the disturbances.

3. Your comparison of the baseline models are somehow obsoleted. There are more recent algorithm about SINDy, Bayesian regression, symbolic learning. To name a few, [1][2][3]. Could you add comparisons to these SOTA model?

[1] Physics-informed learning of governing equations from scarce data.

[2] Bayesian spline learning for equation discovery of nonlinear dynamics with quantified uncertainty

[3] Symbolic physics learner: Discovering governing equations via monte carlo tree search

---

> ### Author Response · Authors · 2024-11-25
>
> $\textbf{W1. Too Much Detail in Appendix.}$
>
> $\textbf{Response:}$
>
> We thank the reviewer for pointing out this issue, and we agree that certain experiment details were overly deferred to the appendix, which may have hindered clarity. To address this, we have moved key details from the appendix to the main paper, particularly in Sections 4.4 to 4.6. Specifically, we now explicitly define the physics-based component $ f $ and the disturbance or residual $ g $ for each experiment in these sections. By including these definitions in the main text, readers can directly understand the setup and the role of ODNN in disentangling $ f $ and $ g $ without needing to reference the appendix.
>
> We hope these changes significantly improve the clarity and accessibility of the experimental details, ensuring that reviewers and readers can follow the methodology and results more easily. Thank you for bringing this to our attention.
>
> $\textbf{W2. Concern about Assumption 1.}$
>
> $\textbf{Response:}$
>
> $\textbf{Acknowledgment of Assumption 1's Limitations:}$ We acknowledge that in many real-world environments, the separation of physics and disturbances cannot always be neatly characterized by properties such as periodicity vs. non-periodicity or monotonicity vs. non-monotonicity. Below, we clarify the scope of our assumptions and discuss their practical applicability.
>
> $\textbf{Scope and Practicality of Assumption 1:}$ Assumption 1 is designed to target a specific subclass of systems where the known physical components and disturbances exhibit distinguishable properties. While this does not cover all real-world scenarios, it applies to many systems where qualitative differences between the components are well-understood. For instance, in power systems, known physics such as power flow equations are often non-periodic, while disturbances caused by environmental factors may exhibit periodic behavior (e.g., daily load cycles). Similarly, in biological systems, monotonic growth models can be distinguished from periodic environmental effects.
>
> The purpose of Assumption 1 is to provide a theoretical framework for disentangling physics and disturbances under idealized conditions. While this assumption may not always hold exactly in real-world environments, it serves as a starting point for designing methods that can approximate this separation in practical settings. For example:
>
> - In some engineering systems, disturbances may exhibit approximate periodicity or convexity, which can be exploited for separation even if the properties are not strictly orthogonal to the physical dynamics.
>
> - In other cases, domain-specific knowledge can be incorporated to define more sophisticated separation criteria beyond simple contrasting properties.
>
>
> $\textbf{Practical Relevance of the Current Work:}$ Despite its limitations, the current work provides a valuable framework for systems where contrasting properties (e.g., convexity vs. non-convexity, monotonicity vs. non-monotonicity) are observed. These properties are common in engineering, physics, and biological systems, making the framework broadly applicable even if not universal. By targeting such systems, ODNN demonstrates how leveraging structural regularization can achieve accurate disentanglement and parameter identification in practical contexts.

---

> ### Author Response · Authors · 2024-11-25
>
> $\textbf{W3. More Experiments Are Needed.}$
>
> $\textbf{Response:}$
>
> We appreciate the reviewer’s observation regarding the scope of experimental cases. Our intention was to focus on carefully controlled scenarios to illustrate the theoretical guarantees and practical performance of ODNN. The synthetic and simplified datasets allow us to validate ODNN's ability to disentangle known physics from unknown disturbances under well-defined conditions, which is crucial for demonstrating the feasibility and robustness of our approach. Meanwhile, we have included two new experiments in the revised manuscript, involving real-world, real-industry systems, demonstrating the applicability of ODNN to complex, realistic scenarios.
>
> $\textbf{Reconstructing Oscillator Dynamics with Observed Data:}$ We consider a real-life temporal dynamic system, the damped harmonic oscillator, initialized with an unknown velocity. This dataset, publicly available from https://www.kaggle.com/datasets/cici118/damped-harmonic-oscillator, simulates the motion of the oscillator governed by the equation $ m x^{''}(t) + c x^{'}(t) + k x(t) = 0 $. The displacement $ x(t) $ is modeled as a combination of known and unknown components, where $ A \cos(\omega t) $ (related to the initial displacement) is regarded as $ f $, and $ B \sin(\omega t) $ (linked to the unknown initial velocity) is treated as $ g $. By leveraging implicit orthogonality, ODNN effectively disentangles these components, recovering the unknown parameter $ B $ with high accuracy. As shown in Figure 5, ODNN accurately reconstructs the dynamics of the oscillator, with the learned $ B $ converging to its true value. This demonstrates ODNN's ability to handle complex, real-world dynamic systems with underlying physical laws.
>
> $\textbf{Balancing Voltage Stability and Power Flow Optimization:}$ We also tackle a real-industry problem in power systems: the Optimal Power Flow (OPF) control problem. This experiment uses the Pecan Street dataset (https://dataport.pecanstreet.org), which provides real-world power flow data from residential and commercial buildings. The objective involves multiple physics functions $ f $, representing the active power flow equations at each bus, and incorporates both convex (voltage stability) and nonconvex (active power generation cost) components. ODNN successfully learns the unknown parameters $ G_{ij} $ and $ B_{ij} $ (conductance and susceptance between buses), recovering the true network topology. This validates ODNN’s scalability to systems involving multiple interconnected physics functions and its applicability to critical operational tasks in power systems.
>
> These new experiments illustrate ODNN's ability to scale to real-world, complex systems in different domains, addressing the concern that previous experiments were limited to toy problems. By handling systems governed by physical laws and involving realistic dynamics, ODNN demonstrates robustness and flexibility in solving practical challenges across diverse applications.

---

> ### Author Response · Authors · 2024-11-25
>
> $\textbf{Q1. Multiple Physics Parameters.}$
>
> $\textbf{Response:}$
>
> We thank the reviewer for highlighting this important point. To address the concern about scaling to systems with multiple physics functions and real-world complexity, we have added a new experiment focusing on a real-industry problem in power systems: the Optimal Power Flow (OPF) control problem. This example demonstrates how ODNN can handle multiple physics functions $ f $, scaling to more complex systems.
>
> In this experiment, the objective is to optimize the trade-off between maintaining voltage stability and minimizing active power generation costs. The formulation involves multiple physics functions $ f $, where each $ f $ corresponds to a power flow measurement, and their summation determines the power injection at each bus. Specifically:
> $
> P_i = \sum_j V_i V_j \left(G_{ij} \cos d\theta + B_{ij} \sin d\theta \right),
> $
> where $ G_{ij} $ and $ B_{ij} $ are the unknown conductance and susceptance between buses $ i $ and $ j $, and $ d\theta $ is the phase angle difference.
>
> Using the Pecan Street dataset (https://www.kaggle.com/datasets/cici118/damped-harmonic-oscillator), we validated the ability of ODNN to recover these parameters in a realistic setting with multiple interconnected nodes. Figure 5 illustrates the learning trajectory of the parameters $ G_{ij} $. The results show that $ G_{ij} $ converges to the true conductance values for connected buses and to zero for non-connected pairs, reflecting the true network topology. A similar level of accuracy is achieved for $ B_{ij} $.
>
> This experiment demonstrates that ODNN can scale to systems involving multiple physics functions by leveraging the distinction between convex and nonconvex components, as well as separable contributions from interconnected nodes. System operators can use the learned parameters $ G $ and $ B $ to restore the network topology and recover hidden physics, enhancing operational insights and system monitoring. The success of ODNN in this experiment provides strong evidence of its applicability to real-world complex systems beyond this specific example.
>
> $\textbf{Q2. Concern About Assumption.}$
>
> $\textbf{Response:}$
>
> We appreciate the reviewer’s insightful comment regarding the challenges of separating physics and disturbances when their behaviors overlap, such as disturbances being proportional to state variables or following similar trends. Below, we address this concern and clarify how ODNN addresses these cases.
>
> $\textbf{Assumptions and Practical Scope}$:
> We acknowledge that the assumption of a clear boundary between physics and disturbances is a limitation in real-world scenarios. ODNN relies on the presence of distinct properties (e.g., periodicity, convexity, or monotonicity) to disentangle these components effectively. While this assumption holds for many practical systems, we recognize that cases where disturbances are proportional to or aligned with physical variables present a more challenging boundary.
>
> $\textbf{Handling Overlapping Trends}$:
> For scenarios where disturbances mimic the trend of physical components, ODNN’s structural regularization framework can still provide meaningful separation by leveraging qualitative differences. For example:
>
> - If the disturbances exhibit proportional relationships but are monotonic while the physical components are non-monotonic, ODNN can exploit this orthogonality.
>
> - Conversely, if disturbances are proportional but non-convex, while physics are convex, ODNN leverages this structural contrast to disentangle the two.
>
> In cases where no such distinguishing properties exist, the current framework would require modifications or domain-specific knowledge to infer additional constraints.
>
> $\textbf{Potential Extensions to Address Overlap}:$ To extend ODNN to systems with overlapping properties between physics and disturbances:
>
> - We could incorporate additional constraints, such as temporal correlations, statistical independence, or domain-driven priors, to enhance separation.
>
> - A potential approach is to adapt the network to focus on latent representations, where disentanglement might be more feasible even when the raw trends overlap.
>
> - Collaborative training with auxiliary datasets, where disturbances and physics are known separately, could help improve the robustness of the separation.
>
> We aim to explore these extensions in future work, broadening ODNN’s applicability to systems with less distinct boundaries. While the current framework addresses a significant subclass of systems with clear separability, we agree that addressing scenarios with ambiguous boundaries is a critical direction for advancing the methodology.

---

> ### Author Response · Authors · 2024-11-25
>
> $\textbf{Q3. More Baseline Methods.}$
>
> $\textbf{Response:}$
>
> We thank the reviewer for pointing out this important concern and for suggesting additional recent algorithms for comparison. To address this, we have expanded our experiments to include three more state-of-the-art (SOTA) baseline methods specifically designed for learning hidden physics:
>
> - $\textbf{PINN-SR:}$ A physics-informed learning method that incorporates sparse regression to discover governing partial differential equations (PDEs) from scarce data [1].
>
> - $\textbf{BSL:}$ A Bayesian spline learner that performs equation discovery while quantifying uncertainty [2].
>
> - $\textbf{SPL:}$ A symbolic physics learner leveraging Monte Carlo Tree Search to identify governing equations [3].
>
> The comparison results are presented in Figure 7 of the revised manuscript, where we evaluate the performance of ODNN against these baselines across multiple datasets. The results demonstrate that, in most datasets, ODNN achieves a lower mean absolute percentage error (MAPE) compared to the SOTA methods. This improvement can likely be attributed to the unique focus of ODNN, which specifically targets systems exhibiting the $f+g$ structure. Unlike the baseline methods, which model the entire system as a single equation or differential equation, ODNN leverages the distinction between $f$ (known physics) and $g$ (disturbances or residuals). This capability enables ODNN to disentangle the underlying physics more accurately and achieve robust parameter recovery, even in the presence of noise or complex system dynamics.
>
> We hope these additional experiments provide a more comprehensive evaluation and demonstrate the distinct advantages of ODNN in recovering hidden physics, addressing the reviewer's concern about outdated baseline comparisons. Thank you for this valuable suggestion.

---

### Official Review · Reviewer_d9Ly · 2024-11-04

**Soundness:** 1
**Presentation:** 3
**Contribution:** 3
**Rating:** 5
**Confidence:** 3

**Summary:**

This paper deals with the identification of parameters in physical systems, by splitting the modeled functions into two components:
- A linear combinations of basis functions $f_i$  (e.g Fourrier basis, or other bases encompassing desirable properties such as symmetry, periodicity, etc)
- A highly expressive deep neural network $g$ to model other less structured and less understood phenomenons, such as observational noise or non-linear effects.

The typical culprit of this approach is that the high expressiveness of the DNN allows overtiffing to the observed data, which prevents parameter identification of the basis coefficients.

Therefore, authors propose to impose explicit orthogonality (in the function space, for the typical L2 inner product) by performing the projection of the DNN (as a function) over the orthogonal of the function space spanned by the $f_i$, like a single-step of Gram-Schmidt process.

Alternatively, they propose to enforce something they call implicit orthogonality, where the inner product that defines orthogonality is replaced by choosing properties that are opposite of each other, such as periodicity/non-periodicity, monotonicity, convexity, etc.

The approach is benchmarked on several datasets and tasks, such as a synthetic task, speech/music extraction from noisy audio records, control tasks on robotic datasets with noisy measurements, and heat source identification, watermark removal.

**Strengths:**

### Impact

The experimental section makes a compelling case in favor of the method, in particular the synthetic task 4.1, that illustrates perfectly the inclination to overfitting of competing methods that over-rely on DNN expressiveness. I appreciate the diversity of scenario tackled by the experimental section.

### Simplicity

The method is not only simple, but also very versatile thanks to the flexibility in the choice of basis $f_i$ or architecture $g$, which once again is illustrated by the diversity of tasks successfully tackled.

**Weaknesses:**

### Soundness

I have a few concerns regarding soundness.

The Theorem 1 is a very strong result, but assumption 1 is even stronger, almost unrealistic, and Theorem 1 is a straighforward consequence of Assumption 1 that does most of the hard work. It is very hard to design a family of functions that would be universal approximator of "everything", but the functional space spanned by the $f_i$. Especially because if $n$ is big, the span $\sum_i\theta_if_i$ is extremely expressive too, for example when $f_i$ are Fourrier basis or Chebychev coefficients.

In Theorem 1, the sentence "network trained via Equation (2) can learn the physical parameter θj correctly" should be avoided. Nothing can be said about optimization dynamics. Therefore, it is best to only speak of the properties of the minimizers of Equation 2, without attempting to capture what "training" could even meaning.

I was not convinced by the proof of Corollary 1, which supposedly implies Assumption 1. The most important part of the proof is in lines 879-881: "However, if hODNN can approximate f (x) to an arbitrary degree of accuracy, it suggests that hODNN(x) must contain components that are aligned with f(x). This alignment contradicts the orthogonality condition ⟨hODNN (x), f (x)⟩ = 0.".  This is not precise enough IMHO.
In this context the sentence "must contain components that are aligned with" is a pleonasm that suggests some sort of circular argument. The confusion is made worse by the fact that the paper confuses function space, functions $f$, and predictions $f(x)$.

Furthermore, Assumption 1 is a statement about *function spaces*, whereas Corollary 1 focuses on a single specific orthogonal projection of Equation (2) that assumes that the true $f$ is known, which means that the $\theta_j$ associated to the $f_j$ are known too. But since optimization of $g$ and $\theta_j$ is simultaneous, I'm not sure what happens. Being orthogonal to a specific $\sum_j \theta_jf_j$ is not the same as being orthogonal to all them for all values of $\theta_j$.

### Implicit orthogonality.

I understand authors intent, but I'm not happy with this terminology and I think it's best to avoid it altogether. The paper deals with inner product and projection onto orthogonal complement, which is the precise definition of orthogonality. However, convexity/non-convexity, periodicity/non-periodicity, monotonicity/non-monotonicity are "contrasting properties" (as authors call them) that **do not** translate into any orthogonality, regardless of the inner product used. I know the term "orthogonality" is sometimes loosely used in informal discussion to describe antagonist properties, but in the context of the paper this increases confusion.

Those properties do not define closed spaces (in the topological sense). For example, for many distances over functions spaces, including the L2 norm, it it possible (and easy) to approximate a symmetric/periodic/monotonic/convex function by a sequence of functions that verify none of these properties. In this context, the Assumption 1 does not apply and the theoretical work is irrelevant.

I believe the link between implicit orthogonality and explicit orthogonality should be discussed more. What is the value of the inner product in practice? Is it close to zero?

### Clarity

I think presentation can be improved at times.

For example, it took me time to parse Proposition 2. It would be better to specify that when using a DNN $g$ that fulfills universal approximation, "*any* set of parameters $\hat \theta_j$" is a solution (emphasis on the *any*).

In Equation (3), every inner product $\langle f(x), g(x)\rangle$ should be replaced by $\langle f, g\rangle$ to not confuse points and functions.

I have other questions detailed below.

**Questions:**

1) How do you evaluate the inner product in l270? Do you discretize the interval? If so, how many points? Is it Riemann integration, or something else?

2) This inner product is defined for $\mathbb{R}\rightarrow\mathbb{R}$. How do you handle multivariate inputs?

3) Can you clarify what are the $f_i$ in general? Are you using the same basis every time, or is it different for each task? If so, can you detail for each task the exact expression of $f_j$? Is there unicity of $\theta_j$ coefficients for each of your use-case?

4) Line 268: "Since the signal and disturbance generally occupy distinct frequency bands" => can you comment on this hypothesis? Beside synthetic task of Sec. 4.1 is it verified anywhere else?

5) Task of Sec 4.5. If there is a fifth heat source, I would expect an algorithm to recover it from observations. What is the rational behind using a basis of symmetric functions for the $f_i$ here?

6) You relied on the canonical L2 inner product. Do you think using other inner products over function space could yield interesting results? e.g inner products over Sobolev space.

---

> ### Author Response · Authors · 2024-11-24
>
> $\textbf{W1. Concerns regarding Assumption 1.}$
>
> $\textbf{Response:}$
>
> We acknowledge that the implementation of implicit orthogonality in ODNN relies on general prior knowledge about the properties of the unknown components (e.g., convexity, periodicity). However, we argue that this reliance is practical and aligned with common practices in physics-guided machine learning. Rather than requiring precise knowledge of the unknown components, ODNN leverages minimal and qualitative characteristics, which are often inferred from domain expertise or exploratory analysis.
>
> For example, in fluid mechanics, turbulent forces are commonly convex, in biological systems, environmental effects exhibit periodicity, and in power systems, disturbances like coupling effects are often monotonic. These broad characteristics of the unknown components are sufficient for guiding the selection of neural network architectures, without requiring explicit parameterization or precise functional forms. The followings are examples. Implicit Orthogonality Case (Contrasting Properties): If $f$ and $g$ possess contrasting properties (e.g., non-convexity vs. convexity), Assumption 1 can be satisfied by ensuring that all $f_i$ share the same property. For example, in the power system dataset:
> $
> f_i(v, \delta) = v_i v_k \left(G_{ik} \cos{\delta_{ik}} + B_{ik} \sin{\delta_{ik}}\right),
> $
> where $v_k$ is the voltage magnitude, $\delta_{ik}$ is the voltage angle difference between nodes $i$ and $k$, and $i \in \mathcal{N}(k)$ denotes the connected node indices. In this expression, each $f_i$ is non-convex, which contrasts with the convex nature of $g$. Therefore, using an input convex neural network (ICNN) for $g$ ensures that the output satisfies Assumption 1, as the network is constrained to only output convex functions. This approach demonstrates that satisfying Assumption 1 in such cases is both practical and achievable.
>
> It is also worth noting that leveraging prior knowledge is a standard practice in physics-guided machine learning. Hybrid models typically rely on first-principle equations, and physics-informed neural networks (PINNs) enforce physical constraints during training. Similarly, ODNN uses qualitative priors about the unknown components to design architectures that ensure accurate disentanglement. Unlike traditional hybrid models, ODNN provides theoretical guarantees for this disentanglement while avoiding overfitting or requiring extensive hyperparameter tuning.
>
> To address the reviewer’s concern about the claim of “unknown” components, we propose revising the manuscript to clarify that the method does not assume the unknown components are entirely unobservable. Instead, we will acknowledge that their general properties are inferred from prior knowledge or data exploration. For example, we could revise the text as follows: “ODNN disentangles known physical components from non-physical disturbances, leveraging general knowledge about the latter’s qualitative properties (e.g., convexity, periodicity). This approach balances prior knowledge with flexibility, making it applicable to a broad range of systems.”
>
> $\textbf{W2. Theorem 1 Description.}$
>
> $\textbf{Response:}$
>
> We thank the reviewer for highlighting this important issue. We agree that the original statement of Theorem 1 could be misleading by implying guarantees about the optimization dynamics or the training process, which are beyond the scope of our theoretical results. To address this, we have revised the statement of Theorem 1 to explicitly focus on the properties of the minimizers of the loss function in Equation (2), avoiding any implication about the behavior of the training process.
>
> Specifically, the revised statement of Theorem 1 is as follows: "Under the conditions of Assumption 1, any minimizer of the loss function in Equation (2) corresponds to the correct physical parameter $\theta_j$."

---

> ### Author Response · Authors · 2024-11-24
>
> $\textbf{W3. Concern about Corollary 1.}$
>
> $\textbf{Response:}$
>
> We thank the reviewer for questions. Below, we address the concerns regarding the precision of the argument, the potential circular reasoning, and the confusion between function spaces, functions, and their evaluations.
>
> $\textbf{Clarifying the Argument in Lines 879-881:}$ We agree that the phrase ``must contain components that are aligned with $f(x)$" lacks precision and may appear to suggest circular reasoning. To address this, we revised the proof as:
> - Suppose $h_{\text{ODNN}}$ is constrained to satisfy the orthogonality condition $\langle h_{\text{ODNN}}, f \rangle = 0$. This implies that $h_{\text{ODNN}}$ lies entirely within the orthogonal complement of the span of $f$ in the function space. If $h_{\text{ODNN}}$ could approximate $f(x)$ arbitrarily well, there would exist some scalar $\alpha \neq 0$ such that $h_{\text{ODNN}} = \alpha f + r$, where $r$ is orthogonal to $f$. However, the orthogonality condition $\langle h_{\text{ODNN}}, f \rangle$ would then be:
> $\langle h_{\text{ODNN}}, f \rangle = \alpha \langle f, f \rangle \neq 0,$ which contradicts the assumption that $\langle h_{\text{ODNN}}, f \rangle = 0$. Therefore, $h_{\text{ODNN}}$ cannot approximate $f(x)$ without violating the orthogonality constraint.
>
> This proof aims at avoiding ambiguity and explicitly ties the orthogonality condition to the impossibility of approximating $f$.
>
> $\textbf{Addressing the Distinction Between Function Space, Functions, and Evaluations:}$ We acknowledge that the paper does not adequately distinguish among elements of the function space, the functions themselves (e.g., $f$), and their evaluations at specific points (e.g., $f(x)$). To address this, we have revised the notation throughout the proof to consistently distinguish between:
> - Functions in the space (e.g., $f$, $g$, $h_{\text{ODNN}}$).
> - Evaluations of these functions at specific points (e.g., $f(x)$, $g(x)$, $h_{\text{ODNN}}(x)$).
>
> $\textbf{W4. Concern about Assumption 1 and Corollary 1.}$
>
> $\textbf{Response:}$
>
> We thank the reviewer for this thoughtful observation. Below, we clarify the distinction between Assumption 1 and Corollary 1, address the concern about simultaneous optimization of $g$ and $\theta_j$, and explain how our approach handles orthogonality in this context.
>
> $\textbf{Distinction Between Assumption 1 and Corollary 1:}$  Assumption 1 is a theoretical statement about the separability of $f$ and $g$ in the function space, relying on orthogonality or contrasting properties. Corollary 1, on the other hand, is derived in the specific setting where the true $f$ is known, and the orthogonal projection onto the space spanned by $\{f_i\}$ can be explicitly computed. This distinction highlights that:
> - Assumption 1 applies in a more general sense, providing the foundation for the theoretical framework.
> - Corollary 1 demonstrates how the orthogonal projection works in practice when $f$ (and thus $\theta_j$) is known.
> - We have revised the text to explicitly discuss this distinction and clarify the roles of Assumption 1 and Corollary 1 in the paper.
>
> $\textbf{On Simultaneous Optimization of $g$ and $\theta_j$:}$ The reviewer is correct that in practice, $g$ and $\theta_j$ are optimized simultaneously, which creates a dependency between these variables. This does not invalidate the orthogonality condition but rather implies that:
> - The orthogonality condition $\langle g, \sum_j \theta_j f_j \rangle = 0$ is imposed iteratively during optimization, based on the current estimates of $\theta_j$.
> - As $\theta_j$ updates during the optimization process, $g$ adjusts to maintain the orthogonality condition relative to the updated $\sum_j \theta_j f_j$. This dynamic process ensures that $g$ remains orthogonal to the instantaneous representation of the physical component.
> - To address this, we will add a discussion in the methodology section, clarifying how simultaneous optimization maintains the orthogonality constraint at each step.
>
> $\textbf{Orthogonality Across All Values of $\theta_j$:}$ The reviewer correctly notes that being orthogonal to a specific $\sum_j \theta_j f_j$ at a given step is not the same as being orthogonal to all possible values of $\theta_j$. In our framework:
> - The orthogonality constraint is imposed relative to the current estimate of $\sum_j \theta_j f_j$ during optimization. This ensures practical disentanglement of $g$ and $f$ at every stage of the learning process.
> - While the orthogonality condition is not globally enforced for all values of $\theta_j$, the iterative nature of optimization ensures that the final model respects the disentanglement principle.
> - We will revise the discussion of Corollary 1 to explicitly state this point, ensuring that the practical implications of the orthogonality condition are clear.

---

> ### Author Response · Authors · 2024-11-24
>
> $\textbf{W5. Concerns regarding Implicit orthogonality}$
>
> $\textbf{Response:}$
>
> We thank the reviewer for raising this insightful question. Our initial intent was to use the term "orthogonality" in a broader sense to unify the concepts of "explicit orthogonality" and "contrasting properties" under a single framework. By doing so, we aimed to provide a cohesive approach that allows users to evaluate whether our method is applicable to their specific system:
> - Explicit orthogonality: The first step is to determine whether $f$ and $g$ are approximately orthogonal. If this is the case, our approach for enforcing explicit orthogonality can be applied.
> - Contrasting property (implicit orthogonality): If $f$ and $g$ are not explicitly orthogonal, the next step is to check whether they possess contrasting properties (e.g., convexity vs. non-convexity, periodicity vs. non-periodicity). In such cases, we refer to it as implicit orthogonality.
>
> We acknowledge that this broader usage of "orthogonality" might lead to confusion, as the term has a strict mathematical definition related to inner products and orthogonal complements. To address this, we will revise as follows:
> - We will use the term "contrasting properties" to describe relationships such as convexity vs. non-convexity or periodicity vs. non-periodicity.
> - The term "orthogonality" will be reserved exclusively for contexts involving inner products and projections, ensuring consistency with its precise mathematical definition.
>
> $\textbf{W6. Closed functional spaces.}$
>
> $\textbf{Response:}$
>
> We thank the reviewer for raising this insightful concern. Below, we address the issue regarding the lack of topological closure of the properties mentioned and clarify the role and relevance of Assumption 1 in our theoretical framework.
>
> We agree with the reviewer that properties such as symmetry, periodicity, monotonicity, and convexity do not define closed spaces in the topological sense. As the reviewer points out, it is indeed possible to approximate a function possessing these properties by a sequence of functions that do not satisfy them. This is a well-known limitation of these properties when viewed through the lens of functional analysis.
>
> Despite the lack of topological closure, Assumption 1 is not intended to rely on the strict topological separation of functions based on these properties. Instead, it serves as an idealized abstraction for cases where $f$ and $g$ are sufficiently distinct in their behavior, either due to orthogonality or contrasting properties. The goal of Assumption 1 is to provide a theoretical framework for disentanglement, rather than to claim universal applicability across all function spaces. For instance:
> - In practical applications, functions with periodic vs. non-periodic or convex vs. non-convex behavior are often distinguishable based on their impact on system dynamics. These distinctions are sufficient for the success of our method, even if the associated function spaces are not topologically closed.
> - The framework relies on the ability to constrain the neural network $h_{\text{ODNN}}$ to output functions that align with $g$ while remaining orthogonal (explicitly or implicitly) to the known physical basis $f$. This practical separability is sufficient to implement the framework, as demonstrated in our experiments.
>
>
> While Assumption 1 does not apply to all function spaces, it remains relevant in scenarios where $f$ and $g$ are distinguishable based on domain knowledge or practical constraints. For example:
> - In the context of signal processing, periodic disturbances can often be separated from non-periodic physical signals using frequency-domain techniques.
> - In optimization problems, convex residual terms are often distinguishable from non-convex physical dynamics due to their differing properties in gradient-based methods.
> - These practical scenarios demonstrate that Assumption 1, though idealized, captures the essence of separability in many real-world systems.

---

> ### Author Response · Authors · 2024-11-24
>
> $\textbf{W7. Relationship between implicit/explicit orthogonality.}$
>
> $\textbf{Response:}$
>
>
> We thank the reviewer for this comment.
>
> - Implicit orthogonality refers to the contrasting properties of $f$ and $g$ (e.g., convexity vs. non-convexity, periodicity vs. non-periodicity) that naturally reduce their overlap in the function space. This is a heuristic or design principle used to model $f$ and $g$ as distinct components.
> - Explicit orthogonality is formally defined by the inner product: $\langle f, g \rangle = \int_\Omega f(x) g(x) \, dx,$ where a near-zero value of $\langle f, g \rangle$ quantitatively confirms orthogonality in the $L_2$ sense.
> - Their relationship: we provide these two orthogonality cases to cover many real-life scenarios where $f$ can be disentangled from $g$. One can first check if $f$ is approximately orthogonal to $g$. If so, our approach for explicit orthogonality can be applied. If not, one can further check if $f$ and $g$ has contrasting properties (e.g., convexity vs. non-convexity, periodicity vs. non-periodicity) and utilize our approach for implicit orthogonality.
>
> $\textbf{Value of the inner product:}$ in practice, the inner product $\langle f, g \rangle$ is used to measure the degree of explicit orthogonality. Below, we provide observations from our experiments:
> - Synthetic dataset (Sec 4.1): Since $f$ and $g$ are deliberately constructed to be orthogonal, the inner product satisfies $|\langle f, g \rangle| < 10^{-3}$, confirming near-complete orthogonality.
> - Audio dataset (Sec 4.2): The noise and audio signal exhibit slight overlap in frequency bands, resulting in a slightly larger inner product, $\langle f, g \rangle \approx 0.02$. Despite this, the method performs well, demonstrating robustness to approximate orthogonality.
> - Other experiments (implicit orthogonality cases): For tasks where implicit orthogonality is applied (e.g., Sec 4.4, power systems), we do not expect $\langle f, g \rangle$ to be near zero. Instead, the contrasting properties of $f$ and $g$ (e.g., periodic vs. non-periodic behavior) ensure sufficient separability for effective performance.
>
> $\textbf{W8. Clarity of of inner product}$
>
> $\textbf{Response:}$
>
> We thank the reviewer for pointing out this issue. To eliminate confusion, we will replace all instances of $\langle f(x), g(x)\rangle$ in Equation (3) and related discussions with $\langle f, g \rangle$, where $\langle f, g \rangle$ explicitly denotes the inner product of functions $f$ and $g$ over the appropriate domain. For instance, Equation (3) will be changed to $h_{\text{ODNN}}(x) = h_{\text{DNN}}(x) - \frac{\langle h_{\text{DNN}}, f \rangle}{\langle f, f \rangle} f(x)$.
>
> $\textbf{Q1-Q2. Calculation of of inner product}$
>
> $\textbf{Response:}$
>
> We thank the reviewer for their detailed reading and for raising this important practical question. Below, we clarify how the inner products are evaluated in our implementation:
> - Evaluation of $\langle f, f \rangle$: since $f$ is a known physics basis function, $\langle f, f \rangle$ is computed using Riemann integration over the domain of $f$. That is, we rely on the analytical form of $f$ to evaluate the integral directly.
> - Evaluation of $\langle h_{\text{DNN}}, f \rangle$: the function $h_{\text{DNN}}$ is not expressed in closed form but is represented by a set of data points generated during training. Therefore, $\langle h_{\text{DNN}}, f \rangle$ is calculated by discretizing the inner product integral into a Riemann summation: $\langle h_{\text{DNN}}, f \rangle \approx \sum_{i=1}^N h_{\text{DNN}}(x_i) f(x_i) \Delta x,$ where $x_i$ are the discretization points, $N$ is the number of data points, and $\Delta x$ represents the spacing between these points.
> -  Number of discretization points: the number of discretization points $N$ corresponds to the number of data points in our experiments. In practice, $N$ is above 1,000 in most of our experiments, ensuring sufficient resolution for accurate evaluation of the inner product.
> - Inner product calculation of multivariate inputs. For multivariate inputs, such as $f(x, y):\mathbb{R}^2 \to \mathbb{R}$ in our synthetic image dataset in Sec 4.1, the inner product is extended naturally by integrating over the multidimensional domain. Specifically, the inner product is defined as: $\langle f, f \rangle = \int_\Omega f(x, y) g(x, y) \, dA,$ where $\Omega \subseteq \mathbb{R}^2$ is the domain of interest, and $dA$ is the infinitesimal area element. For $\langle h_{\text{DNN}}, f \rangle$, this integral is approximated numerically using a Riemann summation. We discretize the domain $\Omega$ into a grid of $N_x \times N_y$ points and compute: $\langle h_{\text{DNN}}, f \rangle \approx \sum_{i=1}^{N_x} \sum_{j=1}^{N_y} f(x_i, y_j) h_{\text{DNN}}(x_i, y_j) \Delta x \Delta y,$ where $(x_i, y_j)$ are the discretized grid points, and $\Delta x$ and $\Delta y$ are the grid spacings along the $x$- and $y$-axes, respectively.

---

> ### Author Response · Authors · 2024-11-24
>
> $\textbf{Q3. What are $f$ in general?}$
>
> $\textbf{Response:}$
>
> We thank the reviewer for this important question and the opportunity to clarify the role of $f_i$, the choice of basis functions across different tasks, and the unicity of $\theta_j$ coefficients.
>
> In general, $f_i$ refers to the physics basis function in the system, which is assumed to be known due to the underlying physical laws of the task. The choice of $f_i$ depends on the specific task since the governing physical laws vary across tasks. The specific expressions for $f_i$ in each task are as follows:
>
> $\textbf{Synthetic Dataset (Sec 4.1)}$:
> - For the 1D case, we choose $f(x) = \sin(4\pi x)$.
> - For the 2D case, we extend this to $f(x, y) = \sin(4\pi x) \sin(4\pi y)$.
>
> $\textbf{Audio Dataset (Sec 4.2)}$:
> - The known noise in the audio is modeled as $f(t) = 0.3\sin(440\pi t) + 0.3\sin(660\pi t) + 0.4\sin(880\pi t)$, representing harmonic components of the noise.
>
> $\textbf{Reinforcement Learning Dataset (Sec 4.3)}$:
> - Here, $f$ is the true reward for upward movement, expressed as $f = \frac{z}{\Delta t}$, where $z$ denotes the post-action $z$-coordinate, and $\Delta t$ is the frame time.
>
> $\textbf{Power System Dataset (Sec 4.4)}$:
> - For each node $i$, $f_k$ represents the power flow equation:
> $
> f_k(v, \delta) = v_i v_k \left(G_{ik} \cos{\delta_{ik}} + B_{ik} \sin{\delta_{ik}}\right),
> $
> where $v_i$ is the voltage magnitude, $\delta_{ik}$ is the voltage angle difference between nodes $i$ and $k$, and $k \in \mathcal{N}(i)$ denotes the connected node indexes.
>
> $\textbf{Heat Transfer Dataset (Sec 4.5)}$:
> - $f_i$ corresponds to the known heat source:
> $
> f_i(x, y, t) = \frac{Q_i}{4 \pi \alpha t} \exp\left( -\frac{(x - x_i)^2 + (y - y_i)^2}{4 \alpha t} \right),
> $
> where $Q_i$ represents the strength of each heat source, $(x_i, y_i)$ are the coordinates of the heat sources, and $\alpha$ is the thermal diffusivity.
>
> $\textbf{Driven Pendulum Dataset (Sec 4.7)}$:
> - The physics basis is $f(t) = \frac{f_0}{2\omega} t\sin(\omega t)$, where $\omega$ is the oscillator frequency, and $f_0$ is the unknown driving force.
>
>
> $\textbf{Biology Growth Dataset (Sec 4.7)}$:
> The known exponential growth is modeled as:
> $
> f(t) = \frac{S}{S+K_S}\exp\left(\frac{S}{K_S}t\right),
> $
> where $K_S$ is the half-saturation constant, and $S$ is the substrate concentration.
>
> Due to space constraints, we have detailed some of the task-specific physics expressions in the corresponding appendix sections.
>
> $\textbf{Unicity of $\theta_j$ Coefficients:}$ For real-world systems, $\theta_j$ coefficients are unique and carry specific physical meanings. For instance, in the Power System Dataset (Sec 4.4), the $\theta_j$ coefficients correspond to $G_{ik}$ and $B_{ik}$, the physical parameters of line $ik$. These values are unique to the IEEE 18-node system and are listed in Figure 14. For synthetic datasets, $\theta_j$ does not have a real-world physical meaning. In these cases, we simulate various values of $\theta$ and evaluate the robustness of the method by calculating the averaged estimation error, as shown in Figure 9.
>
> $\textbf{Q4. Signal and disturbance occupy distinct frequency bands.}$
>
> $\textbf{Response:}$
>
> We thank the reviewer for raising this important question. This hypothesis is based on common practices in signal processing and physical systems modeling. Many real-world systems exhibit this property due to physical constraints or engineering designs. For example:
> - In audio processing, signals of interest (e.g., speech or music) typically occupy lower to mid-frequency bands, while noise (e.g., electrical hum or high-frequency hiss) is often confined to higher frequency bands [Oppenheim1999].
> - In power systems, harmonic disturbances caused by nonlinear loads are well-known to occupy higher frequency bands compared to the fundamental frequency of the system [Farhangi2009].
> - In wireless communication, signals are often designed to occupy specific frequency bands to minimize interference, leaving disturbances or noise in other bands [Proakis2001].
> - These distinct frequency distributions make frequency-based separation a widely applicable and practical assumption in many fields.
>
> $\textbf{References:}$
>
> [Oppenheim1999] Oppenheim A V. Discrete-time signal processing[M]. Pearson Education India, 1999.
>
> [Farhangi2009] Farhangi H. The path of the smart grid[J]. IEEE power and energy magazine, 2009, 8(1): 18-28.
>
> [Proakis2001] Proakis J G. Digital signal processing: principles algorithms and applications[M]. Pearson Education India, 2001.

---

> ### Author Response · Authors · 2024-11-24
>
> $\textbf{Q5. Question of heat source dataset.}$
>
> $\textbf{Response:}$
>
> We thank the reviewer for this insightful question. In the specific setup of Sec 4.5, we assumed that the heat sources are known to be symmetric and spatially fixed at predefined positions. This assumption reflects scenarios where the heat sources represent known entities (e.g., equipment or localized heating elements) with symmetric properties, such as equal intensity or symmetrical placement relative to a central axis. These properties are common in controlled engineering systems or experimental setups.
>
> $\textbf{Q6. Beyond canonical L2 inner products.}$
>
> $\textbf{Response:}$
>
> We thank the reviewer for this insightful question. Indeed, while we rely on the canonical $L_2$ inner product in our current work due to its simplicity and broad applicability, exploring alternative inner products, such as those over Sobolev spaces, could yield interesting results and extensions to our framework.
>
> $\textbf{Canonical $L_2$ Inner Product:}$ The $L_2$ inner product:
> $
> \langle f, g \rangle_{L_2} = \int_\Omega f(x) g(x) \, dx
> $
> is used in this work because it is computationally straightforward, assumes minimal smoothness conditions, and provides a natural measure of similarity for square-integrable functions. However, it is not sensitive to higher-order variations, such as derivatives or spatial gradients, which might be relevant in some applications.
>
> $\textbf{Inner Products Over Sobolev Spaces:}$ Inner products defined over Sobolev spaces, such as:
> $
> \langle f, g \rangle_{H^1} = \int_\Omega f(x) g(x) + \lambda \nabla f(x) \cdot \nabla g(x) \, dx,
> $
> extend the $L_2$ inner product by incorporating terms that account for the derivatives of the functions. These inner products have several potential advantages:
> - They measure not only the alignment of the functions themselves but also the alignment of their gradients or higher-order derivatives, which can capture more complex structural relationships.
> - They are particularly useful in applications where the smoothness or gradient behavior of functions is critical, such as fluid dynamics, elasticity, or image processing.
>
> We also want to mention a few challenges of using Sobolev-based inner products: (1) Evaluating derivatives numerically can increase computational costs, especially for higher-order Sobolev spaces. (2) Estimating gradients from discrete data points can be sensitive to noise and may require sufficient data density to ensure accuracy. Thanks to the reviewer, as a future direction, we plan to explore the incorporation of Sobolev-based inner products into our framework, particularly for tasks involving complex spatial dynamics or higher-order physical laws.

---

> > ### Comment · Reviewer_d9Ly · 2024-11-25
> >
> > Thank you for your lengthy answer.
> >
> > **W1. Concerns regarding Assumption 1.**
> >
> > I think my concern regarding Assumption 1 was misunderstood. I am not questioning the relevance of having prior knowledge about the physical system - quite to the contrary, I agree it is important and frequent.
> >
> > Let me restate what I wrote: "*It is very hard to design a family of functions that would be universal approximator of "everything", but the functional space spanned by the $f_i$.*". I am saying that building a basis $f_i$ and a neural architecture $h_{\text{ODNN}}$ that fulfills Assumption 1 is extremely hard.
> >
> > Having knowledge about your physical system changes nothing about that: how do you even guarantee that a network can represent everything but a convex function / a periodic one / etc? You propose no algorithm to overcome this, since these properties are unrelated to orthogonality and no inner product can be useful here.
> >
> > **On simultaneous optimization of $g$ and $\theta_j$**
> >
> > Do you back-propagate through inner product computation during optimization, or do you treat it as a constant whose value is updated at every iteration?
> >
> > **W6. Closed Functional Spaces.**
> >
> > > The framework relies on the ability to constrain the neural network $h_{\text{ODNN}}$  to output functions that align with $g$ **while remaining orthogonal (explicitly or implicitly)** to the known physical basis $f_i$. This practical separability is sufficient to implement the framework, as demonstrated in our experiments.
> >
> > I don't understand what implicit orthogonality is supposed to mean here. This is simply not something that exists.
> >
> > > In the context of signal processing, periodic disturbances can often be separated from non-periodic physical signals using **frequency-domain techniques**. In optimization problems, convex residual terms are often distinguishable from non-convex physical dynamics due to their differing properties in gradient-based methods. These **practical scenarios** demonstrate that Assumption 1, though idealized, captures the essence of separability in many real-world systems.
> >
> > These examples are not "practical". This is an abstract statement given without references, pretty much of the same kind that ChatGPT would write. What are these "frequency-domain techniques"? What "differing properties in gradient-based methods" ?
> >
> > I am not convinced Assumption 1 is realistic. You provide no evidence you can actually build something fulfilling Assumption 1.
> >
> > **Q1-Q2. Calculation of inner product.**
> >
> > > Therefore, $\langle f, h_{\text{HODNN}}\rangle$ is calculated by discretizing the inner product integral into a Riemann summation
> >
> > What is the value of $\Delta_x$ for your applications?
> >
> > > In practice, $N$ is above 1,000 in most of our experiments, ensuring sufficient resolution for accurate evaluation of the inner product.
> >
> >  Do you have empirical evidence for this? For example error as function of the number of points.
> >
> > > Inner product calculation of multivariate inputs.
> >
> > What is the maximum dimension handled in your experiments? How far do you believe you can go with this approach?

---

### Note · Authors · 2025-01-29

I have read and agree with the venue's withdrawal policy on behalf of myself and my co-authors.